# Tet2 and Tet3 cooperate with B-lineage transcription factors to regulate DNA modification and chromatin accessibility

Chan-Wang Lio[1†], Jiayuan Zhang[2†], Edahí González-Avalos[1], Patrick G Hogan[1], Xing Chang[1,2,3*], Anjana Rao[1,3,4,5*]

[1]Division of Signaling and Gene Expression, San Diego, United States; [2]Key Laboratory of Stem Cell Biology, Institute of Health Sciences, Shanghai Institutes for Biological Sciences, Chinese Academy of Sciences and Shanghai Jiao Tong University School of Medicine, Shanghai, China; [3]Sanford Consortium for Regenerative Medicine, San Diego, United States; [4]Department of Pharmacology, University of California, San Diego, San Diego, United States; [5]Moores Cancer Center, University of California, San Diego, San Diego, United States

**\*For correspondence:**
changxing@sibs.ac.cn (XC); arao@lji.org (AR)

[†]These authors contributed equally to this work

**Competing interests:** The authors declare that no competing interests exist.

**Abstract** Ten-eleven translocation (TET) enzymes oxidize 5-methylcytosine, facilitating DNA demethylation and generating new epigenetic marks. Here we show that concomitant loss of Tet2 and Tet3 in mice at early B cell stage blocked the pro- to pre-B cell transition in the bone marrow, decreased Irf4 expression and impaired the germline transcription and rearrangement of the Igκ locus. Tet2/3-deficient pro-B cells showed increased CpG methylation at the Igκ 3' and distal enhancers that was mimicked by depletion of E2A or PU.1, as well as a global decrease in chromatin accessibility at enhancers. Importantly, re-expression of the Tet2 catalytic domain in Tet2/3-deficient B cells resulted in demethylation of the Igκ enhancers and restored their chromatin accessibility. Our data suggest that TET proteins and lineage-specific transcription factors cooperate to influence chromatin accessibility and Igκ enhancer function by modulating the modification status of DNA.

## Introduction

Cell lineage specification is typically accompanied by changes in DNA cytosine methylation, most of which occur at CpG dinucleotides in somatic cells (*Klose and Bird, 2006*; *Ooi et al., 2009*). This prototypical epigenetic modification, once regarded as static, is now known to be remarkably dynamic (*Pastor et al., 2013*; *Wu and Zhang, 2014*). DNA methyltransferases (DNMTs) convert cytosine to 5-methylcytosine (5mC)(*Ziller et al., 2013*); subsequently, proteins of the TET dioxygenase family oxidise 5mC to 5-hydroxymethylcytosine (5hmC), 5-formylcytosine (5fC) and 5-carboxylcytosine (5caC)(*He et al., 2011*; *Ito et al., 2011*; *Pastor et al., 2013*; *Tahiliani et al., 2009*). These modified bases, together termed oxidized methylcytosines (oxi-mC), facilitate DNA demethylation and are also epigenetic marks in their own right (*Mellén et al., 2012*; *Spruijt et al., 2013*).

Much of the interest in TET proteins has centered around the possibility that oxidised methylcytosines are intermediates in one or more pathways of DNA demethylation (*Pastor et al., 2013*; *Wu and Zhang, 2014*). In one well-documented mechanism, the maintenance DNA methyltransferase complex DNMT1/UHRF1 complex restores symmetrical methylation to the hemi-methylated CpGs that are formed upon DNA replication (*Ooi et al., 2009*). However, 5hmC, 5fC and 5caC all inhibit this process, thus causing 'passive' (replication-dependent) DNA demethylation. In a second mechanism, 5fC and 5caC are substrates for excision by TDG and subsequent 'active' demethylation

through base excision repair (*He et al., 2011*; *Shen et al., 2013*; *Song et al., 2013a*). Thus TET-mediated oxidation of 5mC to 5hmC, 5fC or 5caC facilitates replication-dependent DNA demethylation on the one hand, and replication-independent excision of 5fC and 5caC followed by their replacement with unmodified C on the other hand.

B cells are an essential component of the adaptive immune system, which are selected to expand and subsequently undergo somatic hypermutation and immunoglobulin (Ig) class switching on the basis of their ability to produce high affinity antibodies that provide immunity against pathogens (*Alt et al., 2013*; *Bossen et al., 2012*; *Victora and Nussenzweig, 2012*). B cell development in the bone marrow involves step-wise rearrangement of the genes encoding Ig heavy and light chains, which combine to form the B cell antigen receptor (BCR). The Ig heavy chain locus, Igμ, is rearranged first at the pro-B cell stage. Subsequently, the pre-BCR (formed by pairing the rearranged heavy chain with a surrogate light chain) reactivates Rag expression and stimulates rearrangement of an Ig light chain locus (Igκ or Igλ) at the pre-B cell stage; the Igκ locus predominates, accounting for over 95% of all productively rearranged light chains expressed on B cells in mice. Prior to rearrangement, there are two non-coding germline transcripts from the κ locus, Cκ and Igκ, which are closely correlated with the rearrangement potential of the locus.

As in many other cell types, extensive changes in DNA cytosine modification are observed at a genome-wide level during B cell development (*Kulis et al., 2015*). Specifically, Ig locus rearrangement requires activation of tissue-specific enhancers, a process typically accompanied by changes in DNA and histone modification. In mature B cells, only the rearranged allele of Igκ is demethylated, whereas the unrearranged allele remains methylated. It was thus postulated that methylation suppresses Igκ locus accessibility and that differential methylation may contribute to allelic exclusion; however, subsequent studies found that both κ alleles can be expressed in mature B cells, even though one allele is methylated (*Levin-Klein and Bergman, 2014*). Conversely, analysis of Dnmt1-deficient pre-B cells (which lose DNA methylation at both alleles of the Igκ locus) indicated that demethylation of the Igκ locus was not sufficient to induce locus rearrangement and expression (*Cherry et al., 2000*). Therefore, the relation between DNA methylation and Ig locus rearrangement remains to be fully elucidated at a molecular level.

Here we have examined the role of TET proteins in B cell development. Three TET enzymes, TET1, TET2 and TET3, are expressed in mammalian cells. In mice, Tet1 is highly expressed in the embryo and in embryonic stem cells (*Kang et al., 2015*; *Koh et al., 2011*), whereas Tet2 and Tet3 are abundantly expressed in somatic cells including hematopoietic cells. Using mice doubly deficient for Tet2 and Tet3 in early B cells, we show that TET function is required for developing B cells to transit from the pro-B to the pre-B stage. Tet2 and Tet3 regulate germline transcription and rearrangement of the Igκ light chain, and synergize with B lineage-specific transcription factors such as E2A and PU.1 to promote DNA demethylation and chromatin accessibility at B cell enhancers. TET catalytic activity is required for IRF4 expression, but IRF4 cannot restore Igκ germline transcription or the demethylation status of the Igκ enhancers in the absence of Tet2 and Tet3. Our results suggest that TET enzymes regulate multiple aspects of B cell development at the pro-B to pre-B cell transition, and demonstrate a causal relationship between DNA modification and chromatin accessibility.

## Results

### Loss of TET function in B cells impairs B cell development at the pro-B to pre-B transition

*Tet2* and *Tet3* mRNAs are abundantly expressed at all stages of B cell development, whereas *Tet1* mRNA is expressed at much lower levels (*Ko et al., 2010*)(*Figure 1—figure supplement 1A*, *left*, *blue bars*). However neither *Tet2*$^{-/-}$ mice (which are fully viable and fertile [*Ko et al., 2011*]) nor *Tet3*$^{fl/fl}$ *Mb1Cre* mice (which we generated to bypass the perinatal lethality of *Tet3-/-* mice [*Gu et al., 2011*]) displayed any striking B cell phenotypes (*Figure 1—figure supplement 1B, C and D* and *not shown*). To assess the effect of a profound loss of TET function in B cells, we generated *Tet2*$^{-/-}$ *Tet3*$^{fl/fl}$ *Mb1Cre* mice (here termed *Tet2/3 DKO* mice), in which a conditional *Tet3* allele (*Ko et al., 2015*) is deleted in the context of a germline deletion of *Tet2* at the transition from pre-pro B cells to pro-B cells (*Hobeika et al., 2006*). As judged by DNA dot blot using an anti-5hmC

antibody, 5hmC levels were at least 4-fold lower in vitro-cultured pro-B cells of *Tet2/3* DKO mice compared to wild type (WT) (*Figure 1—figure supplement 1A*, right).

*Tet2/3* DKO mice showed a striking reduction in the percentages and numbers of B cells in the bone marrow compared to WT mice, with a partial block at the pro-B to pre-B transition (*Figure 1*). The percentage of B220$^+$CD19$^+$ cells in the *Tet2/3* DKO bone marrow was substantially reduced (<50% of that in WT bone marrow) at 7–8 weeks and even more pronounced (<10%) at 11–12 weeks of age (*Figure 1A*). The percentages and numbers of pre-B cells (CD43$^{low}$B220$^+$IgM$^-$) and immature B cells (CD43$^{low}$B220$^+$IgM$^+$) in the *Tet2/3* DKO bone marrow at 11–12 weeks were 7–20% of those in the WT bone marrow (*Figure 1B–D*); concomitantly, the percentages and numbers of re-circulating (mature) IgM$^+$IgD$^+$CD19$^+$ B cells in the bone marrow were also greatly diminished in *Tet2/3* DKO mice (*Figure 1C,D*). Because B220 and CD43 are co-expressed not only on B cells but also on plasmacytoid dendritic cells, we reanalyzed CD19$^+$B220$^+$ bone marrow cells based on c-kit and CD25 expression; this analysis confirmed that percentages and numbers of pre-B cell (IgM$^-$CD19$^+$-B220$^+$ckit$^-$CD25$^+$) were substantially reduced in *Tet2/3* DKO mice (*Figure 1E*). In parallel, *Tet2/3* DKO mice showed an increased percentage of pro-B cells (IgM$^-$CD19$^+$B220$^+$ckit$^+$CD25$^-$) in the bone marrow (*Figure 1E*, left), but total pro-B cell numbers were unaltered because of the overall decrease in total B-lineage cells (*Figure 1E*, right). Consistent with these findings, there was a reduction in the percentage and number of mature B cells in the spleen (*Figure 1F*).

A large fraction of CD19$^+$B220$^+$ B cells in the spleen of *Tet2/3* DKO mice lacked cell-surface IgM and/or IgD expression (~25% and~45% IgM$^-$IgD$^-$ cells in eight week-old and 11 week-old *Tet2/3* DKO mice respectively; *Figure 1G*) These peripheral Ig-negative B cells expressed a significantly higher level of Terminal deoxynucleotidyl transferase (TdT) and pre-BCR (VpreB, also known as CD179α), thus displaying the expression profile of developing pro-B cells (*Figure 1H*); they also uniformly lacked expression of *Tet3* mRNA, indicating complete deletion of the *Tet3* allele (*Figure 1—figure supplement 1E*). In contrast, two out of 3 surface Ig-positive cell samples analyzed showed residual expression of *Tet3* mRNA (*Figure 1—figure supplement 1E*). Together, these data suggest that the surface Ig-positive cells in *Tet2/3* DKO mice were 'escapees' that had not completely deleted the *Tet3* allele, and hence had expanded due to the proliferative advantage of B cells expressing a cell-surface B cell receptor (BCR) (*Kraus et al., 2004*).

Notably, there was a detectable expansion of CD11b$^+$ myeloid-lineage cells in the bone marrow and spleen of *Tet2/3* DKO mice (*Figure 1—figure supplement 1F*), resembling the myeloid skewing observed in mice deficient in Tet2 or Tet3 alone in the hematopoietic compartment (*Ko et al., 2011*, *2015*). Moreover, all older *Tet2/3* DKO mice developed B cell lymphomas with splenomegaly and lymphadenopathy by the time they were five months old (*Figure 1—figure supplement 1G*). These findings are reported here for completeness, but will be pursued in a separate study. The data recall the late B cell malignancies observed in germline Tet1-deficient mice (*Cimmino et al., 2015*), and reinforce the notion that TET loss-of-function is associated with hematopoietic malignancies (*Huang and Rao, 2014*; *Ko et al., 2015*).

## Tet2 and Tet3 regulate κ chain expression and the DNA modification status of Igκ locus enhancers in vivo

The splenic *Tet2/3* DKO B cells that lacked cell-surface IgM and/or IgD expression (*Figure 1G*) also showed greatly reduced expression of cell-surface and intracellular Igκ light chains (*Figure 2A*) and impaired Igκ rearrangement (*Figure 2B*). Furthermore, purified pre-B cells from *Tet2/3* DKO mice showed a marked reduction (>50%) in Vκ-Jκ1 rearrangement compared to their WT counterparts (*Figure 2C*), demonstrating that TET function is important for Igκ locus rearrangement and κ chain expression in developing B cells and hence critical for the generation of mature B cells with functional B cell receptors.

TET proteins are known to facilitate both active and passive DNA demethylation at gene bodies, promoter-TSS regions and enhancers ([*Hon et al., 2014*; *Pastor et al., 2011*; *2013*; *Tsagaratou et al., 2016*] and *below*). To ask whether loss of TET function in *Tet2/3* DKO B cells affected Igκ expression through changes in DNA modification, we analyzed the DNA modification status of known *cis*-regulatory elements in the Igκ locus (*Inlay et al., 2002*; *Zhou et al., 2010*) (*Figure 3A*, *Figure 3—figure supplement 1A*). The 3' enhancer (3'Eκ) and a more distal 3' enhancer (dEκ) contain 2 and 3 CpG dinucleotides respectively; the intronic Eκ enhancer (iEκ) contains no CpGs and was not considered further (*Figure 3A*, *Figure 3—figure supplement 1A*). Using bisulfite

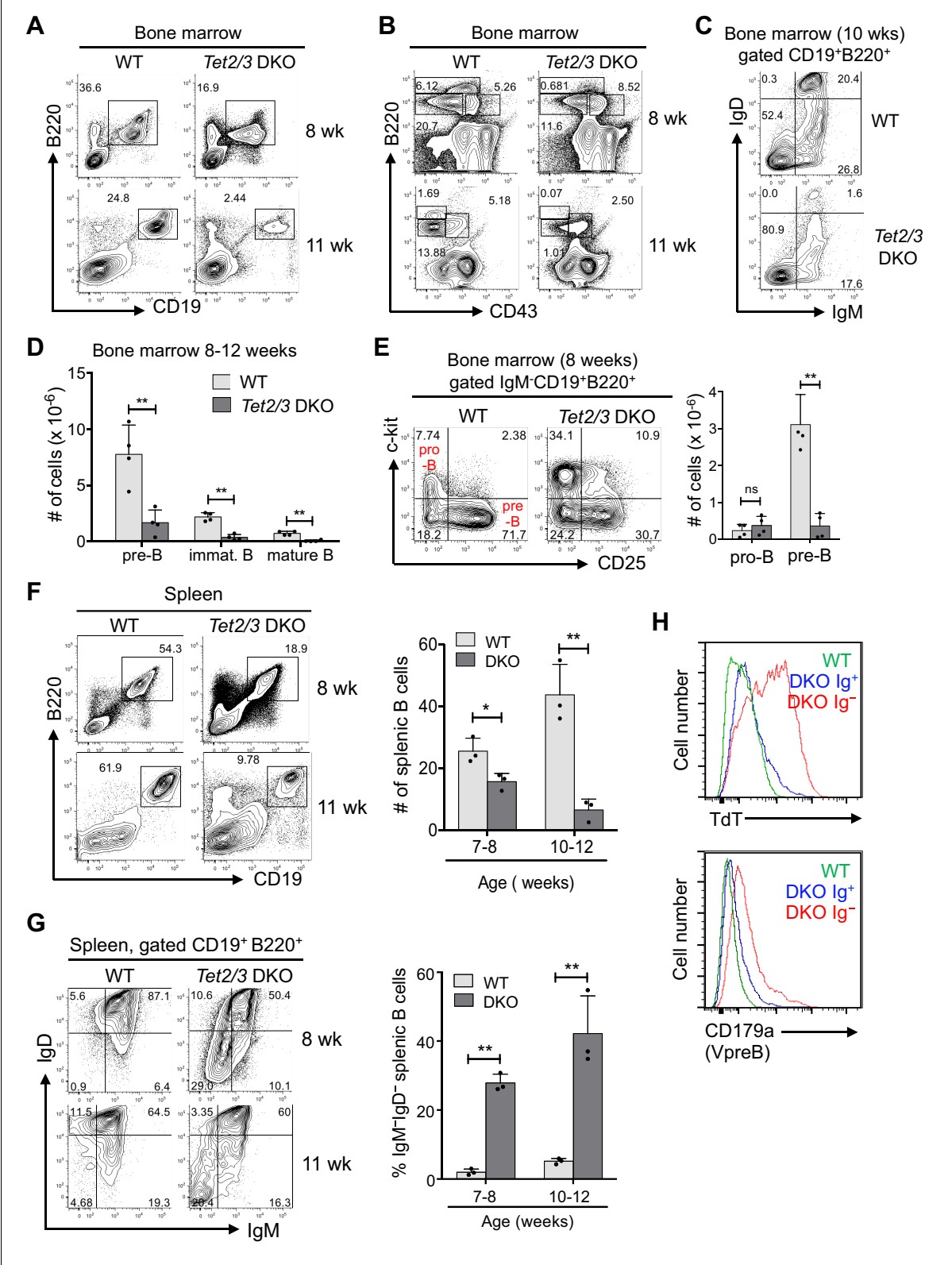

**Figure 1.** Loss of Tet2 and Tet3 in the B cell lineage results in B cell developmental blockade in vivo. (**A**) Reduced bone marrow B cells (B220[+] CD19[+]) in *Tet2[-/-]Tet3[fl/fl] Mb1Cre* (*Tet2/3* DKO) mice. Total bone marrow cells from wild type (WT) or *Tet2/3* DKO mice at eight weeks (upper) and 11 weeks (lower) were analyzed for the percentage of total B cells (CD19[+]B220[+]) by flow cytometry and the representative plots are shown. Note that the loss of B cells is apparent at 8 weeks and more pronounced at 11 weeks. (**B**) *Tet2/3* DKO mice display a striking reduction in pre-B cells. Bone marrow pre-B

*Figure 1 continued on next page*

Figure 1 continued

cells (IgM⁻CD43⁻B220⁺) were analyzed by flow cytometry and representative plots are shown. The absolute numbers of pre-B cells in 8–12 week-old mice are shown in *Figure 1D*. (**C**) Reduced frequency of immature (IgM⁺IgD⁻) and mature recirculating (IgM⁺IgD⁺) B cells in DKO bone marrow. CD19⁺B220⁺ bone marrow cells from 10 week-old *Tet2/3* DKO and WT mice were analyzed for cell surface IgM and IgD expression. Data shown are representative and numbers of immature and mature B cells from four mice are shown in *Figure 1D*. (**D**) Quantification of cell numbers in multiple experiments similar to those shown in *Figure 1B and C*. (**E**) B cell development in *Tet2/3* DKO mice is blocked at the transition from the pro-B cell to the pre-B cell stage. Bone marrow B cells (CD19⁺B220⁺) of *Tet2/3* DKO or WT mice (eight weeks) were analyzed by flow cytometry for c-kit and CD25 expression. *Left panel*, Representative flow cytometry plots; *right panel*, numbers of bone marrow B cells at pro-B and pre-B cell stages were compared between WT and *Tet2/3* DKO mice (n = 4). Pro-B cells, CD19⁺B220⁺IgM⁻CD25⁻c-kit⁺; pre-B cells, CD19⁺B220⁺IgM⁻CD25⁺ c-kit⁻. (**F**) *Tet2/3* DKO mice have reduced splenic B cells. Total splenocytes from WT or *Tet2/3 DKO* mice were analyzed for B220 and CD19 expression by flow cytometry. *Left panel*, representative flow cytometry plots. *Right panel*, numbers of splenic B cells in WT and *Tet2/3* DKO mice (n = 3 for each age group). (**G**) Accumulation of B cells lacking surface IgM or IgD expression in the periphery of *Tet2/3* DKO mice. B cells (CD19⁺B220⁺) were analyzed for IgM and IgD expression. *Left panel*, representative flow cytometry plots. *Right panel*, compiled data of IgM⁻IgD⁻ % from multiple experiments. *Tet2/3* DKO mice show a considerable age-dependent increase in the proportion of IgM⁻IgD⁻ B cells lacking surface B cell receptor. (**H**) IgM⁻IgD⁻ B cells in the Tet2/3 DKO mice expressed high level of TdT (upper panel) and preBCR (CD179a, lower panel). Ages of mice are shown on the right for (**A**), (**B**), (**F**), and (**G**). Error bars indicate standard deviations. *, p<0.05, **, p<0.01 by Student's *t* test. ns, not significant.

The following figure supplement is available for figure 1:

**Figure supplement 1.** Tet2 and Tet3 are redundantly required for B cell development and BCR expression.

sequencing (which does not distinguish 5hmC from 5mC, or 5fC and 5caC from unmodified C [*Huang et al., 2010*; *Pastor et al., 2013*]), we found that all 5 CpG sites in the 3' and distal Eκ enhancers were fully 'methylated' (5mC + 5 hmC) in LSK (Lin⁻, Sca1⁺ c-kit⁺) cells, a population enriched in hematopoietic stem/precursor (HSPC) cells (*Hodges et al., 2011*); began to lose this modification at the pre-pro-B cell stage; were partially methylated in pro-B cells; and almost completely 'unmethylated' (C/5fC/5caC) in WT splenic CD19⁺ B cells (*Figure 3—figure supplement 1B*). Examination of publicly available data for human HSC, neutrophils and B cells showed that the corresponding human enhancers also undergo B cell-specific 'demethylation' (loss of 5mC/5hmC; *Figure 3—figure supplement 1C*).

## Tet2 and Tet3 regulate demethylation of Igκ locus enhancers and Igκ germline transcription *in vitro*

To analyze the defects in *Tet2/3* DKO B cells at a molecular level, we used a well-characterized cell culture system (*Vieira and Cumano, 2004*) in which TET loss-of-function could be acutely induced. *Tet2⁻/⁻Tet3^{fl/fl}* bone marrow cells were cultured for one week with IL-7 and OP-9 feeder cells to generate pro-B cells (also known as large pre-B or pre-B II cells), then retrovirally transduced with Cre-IRES-GFP or control empty IRES-GFP retrovirus to delete the floxed *Tet3* allele (*Figure 3B*). The efficiency of *Tet3* deletion was >90%, and there was only a slight compensatory upregulation of *Tet1* mRNA (*Figure 1—figure supplement 1A*, *left*, *red and green bars*). Under these conditions, expression of surface markers in Cre-transduced *Tet2⁻/⁻Tet3^{fl/fl}* pro-B cells (termed *Tet2/3* DKO pro-B cells) was indistinguishable from those in control cells (WT cells transduced with Cre or *Tet2⁻/⁻Tet3^{fl/fl}* cells transduced with empty vector), with all genotypes exhibiting the typical surface phenotype of pro-B cells (B220⁺, CD19⁺, CD43^{high}, IgM⁻, CD127⁺; *Figure 3—figure supplement 2A* and *not shown*).

Bisulfite sequencing showed that in vitro-generated *Tet2/3* DKO pro-B cells displayed a striking increase in DNA modification (5mC + 5 hmC) at the 3' and distal Igκ enhancers and a parallel decrease (>90%) in Cκ transcripts and Igκ germline (GL) transcripts compared to control pro-B cells (*Figure 3C and D*, *compare blue and green bars*), consistent with the corresponding decreased Igκ expression and increased DNA modification (5mC + 5 hmC) observed in *Tet2/3* DKO B cells compared to WT B cells analyzed ex vivo (*Figures 2A* and *3E*). The increased modification indeed reflected increased DNA methylation (5mC, not 5hmC), since genome-wide 5hmC mapping by CMS-IP showed almost no remaining 5hmC in *Tet2/3* DKO pro-B cells compared to WT pro-B cells (*Figure 3—figure supplement 2B*); this is a more reliable technique in terms of signal-to-noise ratio than DNA dot blot analysis with anti-5hmC (*Figure 1—figure supplement 1A*, *right*). In vitro-generated pro-B cells deficient in Tet2 alone (*Tet2⁻/⁻Tet3^{fl/fl}* cells transduced with empty retrovirus, *Figure 3C-3E*, *red bars*) showed a partial increase in methylation (*Figure 3C*) and a partial reduction

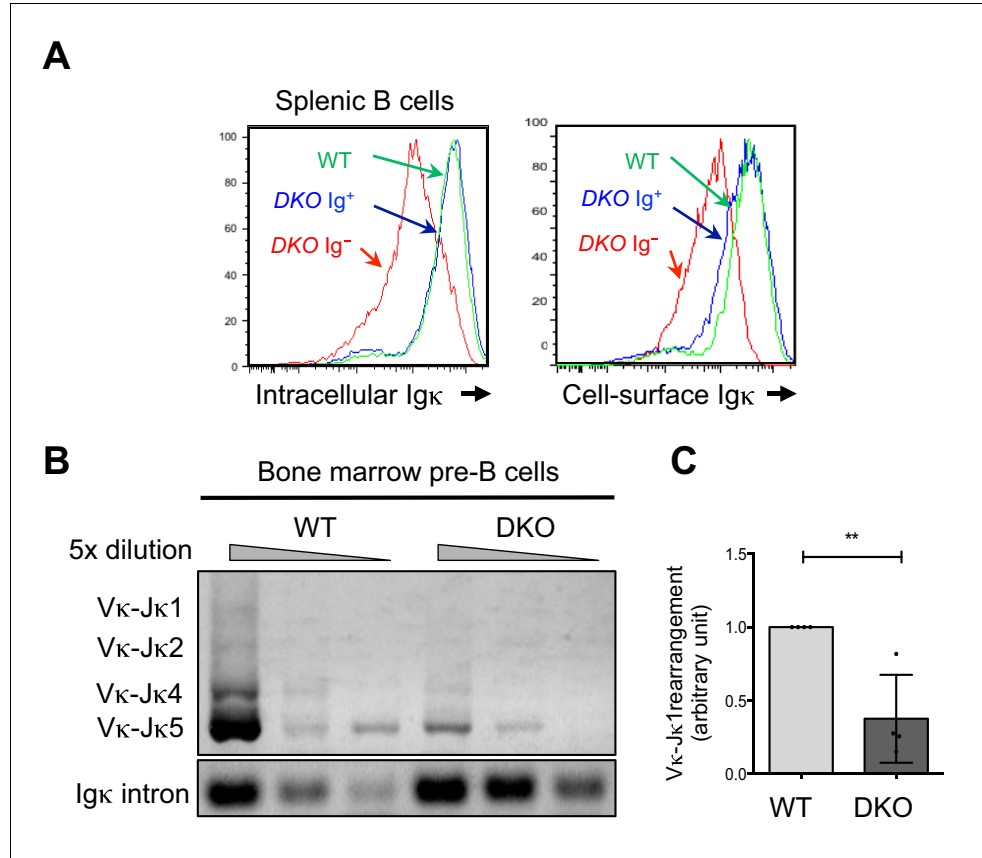

**Figure 2.** Tet2 and Tet3 promote immunoglobulin chain expression and rearrangement in vivo. (**A**) IgM⁻IgD⁻ B cells in the *Tet2/3* DKO mice have diminished immunoglobulin light chain expression. IgM⁻IgD⁻ and IgM⁺IgD⁺ splenic B cells (CD19⁺B220⁺) were analyzed for intracellular (*left panel*) and cell surface (*right panel*) Igκ expression by flow cytometry. Data are representative of three independent experiments. (**B**) *Tet2/3* DKO pre-B cells have reduced Igκ rearrangement. WT and *Tet2/3* DKO pre-B cells (CD19⁺ckit⁻CD25⁺) were isolated from bone marrow by cell sorting and the Igκ rearrangement was determined by PCR amplification with Igκ intron as loading control. Representative experiment of three is shown. (**C**) Vκ-Jκ1 rearrangement is impaired in the *Tet2/3 DKO* pre-B cells. Rearrangement of Vκ-Jκ1 was quantified by real-time PCR. Data are summary of four pairs of mice and were normalized to the signal from WT. **, p<0.01 by Student's *t* test.

in Cκ and Igκ germline transcripts (*Figure 3D*) whereas loss of Tet2 in vivo had no apparent effect (*Figure 3E*), suggesting redundant functions of Tet2 and Tet3 in modulating the DNA modification status of these Igκ enhancers and showing that Tet3 can compensate fully for lack of Tet2 in vivo but not in vitro.

Igκ germline transcription and Igκ rearrangement is regulated by the concerted action of multiple transcription factors at the Eκ enhancers, including E2A, EBF1, PAX5, PU.1, IRF4 and IRF8 (*Clark et al., 2014*; *Cobaleda et al., 2007*; *Hagman and Lukin, 2006*; *Murre, 2005*)(*Figure 3A*). With the exception of the functionally redundant transcription factors IRF4 and IRF8 which showed decreased expression (*see below*), mRNA and protein levels for most of these known regulators were unchanged in *Tet2/3* DKO pro-B cells compared to WT (*Figure 3D*, *Figure 3—figure supplement 2C* and *not shown*). Similarly, *Tet2/3* DKO pre-B cells transformed with BCR-ABL (*Chen et al., 1994*; *Klug et al., 1994*) had diminished Igκ germline transcription, both under resting conditions and after treatment with the Abelson kinase inhibitor Gleevec (also known as imatinib or STI-571), which induces Igκ locus activation in BCR-Abl-transformed cells (*Muljo and Schlissel, 2003*) (*Figure 3—figure supplement 2D*). Gleevec-treated *Tet2/3* DKO pre-B cells also had significantly reduced formation of Rag-induced DNA double-strand breaks at the Jκ1 segment compared to WT

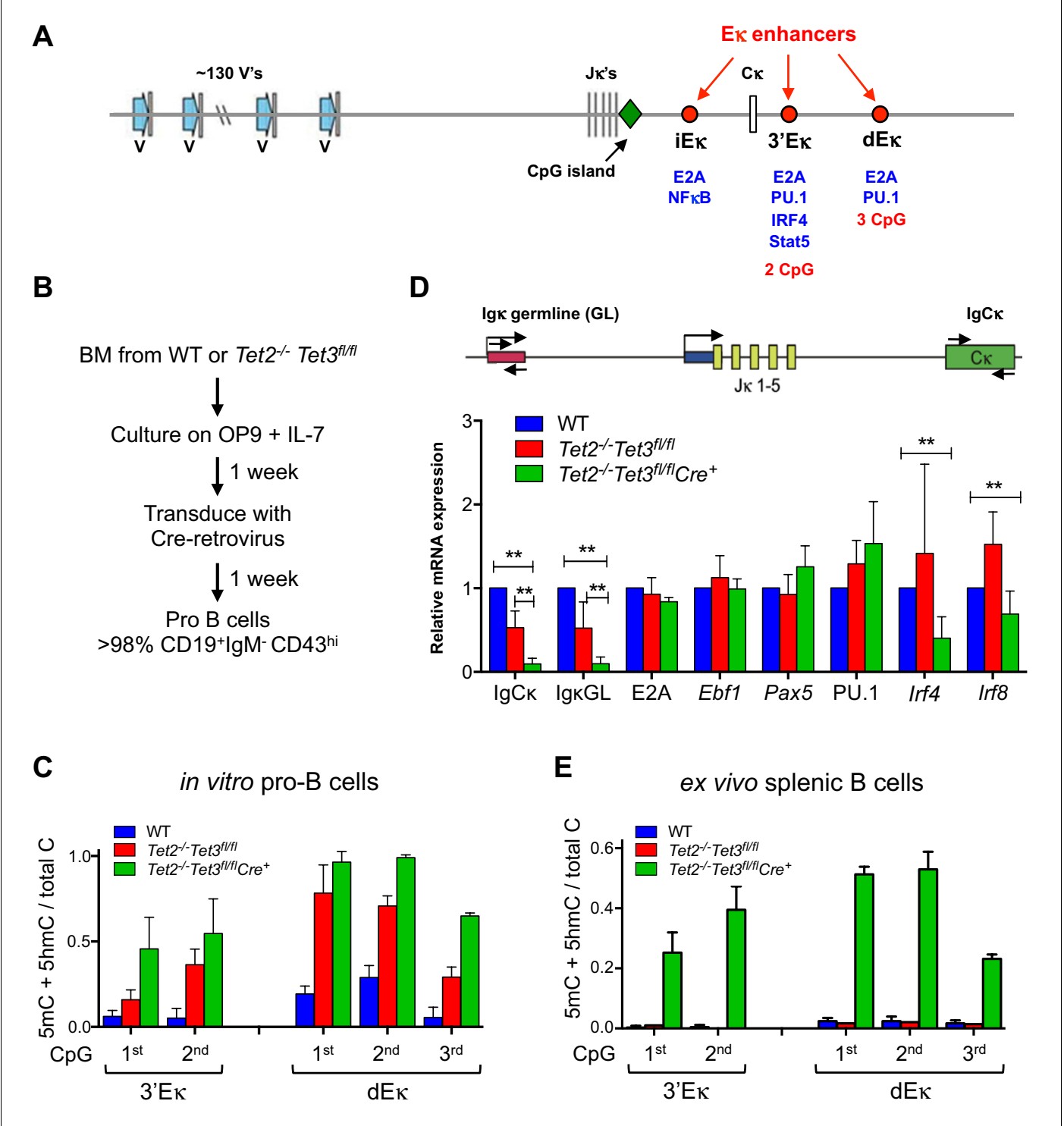

**Figure 3.** Tet2 and Tet3 promote germline transcription of the Igκ locus and demethylation of the 3' and distal Eκ enhancers. (**A**) The Igκ locus. Diagram depicts gene segments and regulatory elements of the Igκ locus. For sequences of the 3' and distal Eκ enhancers, see *Figure 3—figure supplement 1A*. (**B**) Flow-chart depicting generation of *Tet2/3* DKO pro-B cells in vitro. Bone marrow (BM) cells from wild type (WT) or *Tet2⁻/⁻Tet3fl/fl* mice were cultured on OP9 stromal cells with IL-7 (10 ng/ml) for one week, then transduced with Cre-IRES-GFP retrovirus. GFP⁺ cells were isolated by cell sorting five days later, and cultured on OP9 cells with IL-7 for an additional week. At the end of this period, all cells expressed similar surface markers (B220⁺CD43⁺IgM⁻), consistent with the phenotypes of pro-B cells (*Figure 3—figure supplement 2A*). *Tet3* deletion efficiency was >90% (not shown). (**C**) Acute loss of TET function results in increased DNA 'methylation' (5mC + 5 hmC) at the 3' and distal Eκ enhancers. In vitro-derived pro-B cells from WT, *Tet2* KO and *Tet2/3* DKO mice were analyzed for the DNA modification status of 3' and distal Eκ enhancers by bisulfite treatment of

*Figure 3 continued on next page*

*Figure 3 continued*

genomic DNA followed by PCR amplification and sequencing on an Illumina MiSeq platform. Error bars show the standard deviation of three independent experiments. (**D**) Tet2 and Tet3 are required for germline transcription of the Igκ locus and *Irf4/8* in pro-B cells. *Top*, Diagram of the Igκ locus and the primers used to detect germline and Cκ transcription. *Bottom*, WT, *Tet2* KO and *Tet2/3* DKO pro-B cells were analyzed by real time PCR for expression of indicated genes. mRNA expression was normalized to that of *Actb*, and mRNA levels in WT cells were set to 1. Note the almost complete absence of germline and Cκ transcription in *Tet2/3* DKO pro-B cells. Error bars represent the standard deviation of three independent experiments. \*\*, p<0.01 in Student's *t* test. (**E**) Loss of TET function in vivo is accompanied by increased DNA 'methylation' (5mC + 5 hmC) at the 3' and distal Eκ enhancers. CD19$^+$ cells were isolated from spleens of WT, Tet2KO (*Tet2$^{-/-}$Tet3$^{fl/fl}$*) and *Tet2/3* DKO (*Tet2$^{-/-}$Tet3$^{fl/fl}$ Mb1Cre*) mice, and the DNA modification status of the 3' and distal Eκ enhancers was determined as in (**C**). Error bars indicate the range of values obtained in two independent experiments.

The following figure supplements are available for figure 3:

**Figure supplement 1.** The 3'Eκ and distal Eκ enhancers undergo B cell specific-loss of cytosine modification.

**Figure supplement 2.** Tet2/3 regulate Igκ expression and rearrangement in BCR-Abl transformed pre-B cells.

cells (*Figure 3—figure supplement 2E*). These results show that TET proteins demethylate and modulate the activity of Igκ enhancers and thus influence subsequent Igκ rearrangement.

## TET proteins promote chromatin accessibility at binding sites for B cell lineage-specific transcription factors

Ig germline transcription requires locus accessibility (*Yancopoulos and Alt, 1985*). To test whether TET proteins had a role in regulating the accessibility of the Igκ locus, we identified accessible chromatin regions in cultured pro-B cells at a genome-wide level by ATAC-seq (assay for transposase-accessible chromatin using sequencing) (*Buenrostro et al., 2013*) (*Figure 4*). The majority of accessible regions (n = 44668, 79.8%; see *Figure 4* legend for details) were 'commonly accessible', i.e. similarly accessible in WT compared to *Tet2/3* DKO pro-B cells (*Figure 4A*, *grey dots*). Importantly, 1303 regions (2.2%) were potentially regulated by TET because they were more accessible in WT than in *Tet2/3* DKO pro-B cells (*Figure 4A*, WT>DKO differentially accessible regions (WT>DKO DARs); *blue dots*). 1192 regions (2.1%) gained accessibility in the *Tet2/3* DKO cells (*Figure 4A*, DKO>WT differentially accessible regions (DKO>WT DARs); *red dots*), and are likely affected indirectly by TET (*see below*).

To correlate TET activity and chromatin accessibility, we analyzed genome-wide 5hmC distributed by CMS-IP (*Figure 4B*). Compared to randomly sampled regions (*Figure 4B*, *dotted line*), all accessible regions displayed higher levels of 5hmC (*solid lines*), suggesting an overall positive correlation between TET activity (5hmC deposition) and chromatin accessibility. DARs that were more accessible in WT than in DKO pro-B cells (TET-regulated DARs, WT>DKO DARs or simply, WT DARs) exhibited significantly higher levels of 5hmC compared to other accessible regions (*Figure 4B*), whereas commonly accessible regions and DARs that were more accessible in DKO than in WT (DKO>WT DARs or simply, DKO DARs) had lower, comparable levels of 5hmC (*Figure 4B*, *compare red and grey lines*), further supporting the notion that TET activity is required to maintain chromatin accessibility. The majority (>86%) of TET-regulated DARs exhibit the characteristics of enhancers: they are located distal to transcription start sites (*Figure 4C*) and are enriched for H3K4me1 over H3K4me3 (*Creyghton et al., 2010*; *Shlyueva et al., 2014*) (*Figure 4D*). As expected from this global analysis, TET deficiency greatly decreased the accessibility of the dEκ enhancer (*Figure 4E*).

We mined the differentially accessible regions for consensus transcription factor binding motifs, identified by motif enrichment analysis (*Heinz et al., 2010*) and ChIP-seq. Comparing TET-regulated regions (WT>DKO DARs) to commonly accessible regions, the E-box/ basic region-helix-loop-helix (bHLH) (CANNTG) motif was highly represented, with the second most prominent motif being the composite ETS:IRF (PU.1-IRF) motif (*Figure 4—figure supplement 1A*). Consistent with their enrichment for bHLH and ETS binding motifs, WT>DKO DARs showed strong enrichment for PU.1 and E2A binding sites identified experimentally by ChIP-seq (*Heinz et al., 2010*; *Lin et al., 2010*), compared to DARs more accessible in DKO than in WT, commonly accessible regions and control random regions (*Figure 4F,G*) In contrast, the distribution of binding sites for other relevant

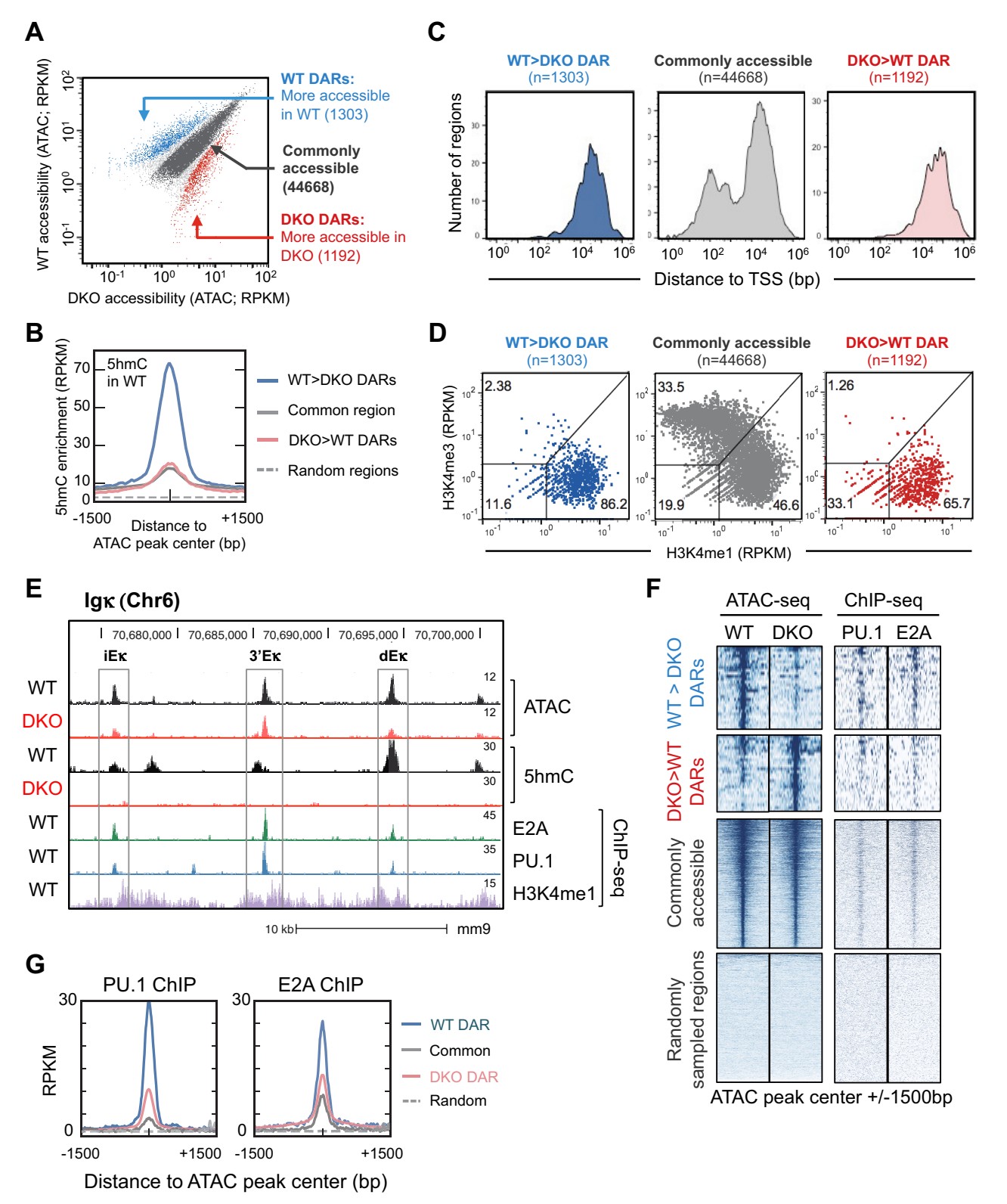

**Figure 4.** Tet2 and Tet3 regulate genome-wide enhancer accessibility in B cells. (A) ATAC-seq identifies TET-regulated accessible regions in the genome. WT and *Tet2/3* DKO pro-B cells were generated as described in *Figure 3* and used to prepare ATAC-seq libraries with two replicates for each genotype. Accessible regions were identified by MACS2 and reads enrichment at each region was complied by MEDIPS and is shown as mean Reads Per Kilobase per Million mapped reads (RPKM) from two replicates, with each dot represent one accessible region (total region = 55999). Differential

*Figure 4 continued on next page*

*Figure 4 continued*

accessible regions were identified by MEDIPS with an adjusted *p* value ≤ 0.1. Blue dots indicate regions with higher accessibility in WT, or WT differential accessible regions (WT DAR; n = 1303); red dots indicate regions with higher accessibility in DKO, or DKO DAR (n = 1192); grey dots indicate regions with no statistically significant (p>0.1) with darker grey indicating regions with less than 4-fold difference (n = 44668) which was used for subsequent analysis as commonly accessible regions. (**B**) 5hmC is highly enriched in WT DARs. Genome-wide distribution of 5hmC in WT pro-B cells was assessed by CMS-IP (*Figure 3—figure supplement 2B*) and the enrichment of 5hmC across indicated accessible regions (centered and extended ± 1500 bp) was plotted with y-axis showing the mean RPKM. Random genomic regions (n = 44668) were shown as reference for background level of 5hmC. (**C**) TET-regulated WT DARs are primarily distal to transcription start sites (TSS). Distance between regions from (**A**) and closest TSS was plotted on X-axis (log 10) with a number of regions on Y-axis. (**D**) Tet2 and Tet3 regulate the accessibility of enhancers in pro-B cells. The enrichment of H3K4me1 and H3K4me3 (*Lin et al., 2010*) at regions surrounding ATAC-seq peaks (±250 bp) were analyzed and plotted, with X-axis indicating H3K4me1 and Y-axis indicating H3K4me3. ATAC-seq peaks were classified into promoters (H3K4me1$^{low}$, H3K4me3$^{high}$), enhancers (H3K4me1$^{high}$, H3K4me3$^{low}$), or other regions (H3K4me1$^{low}$, H3K4me3$^{low}$), which may include insulators, silencers and locus control regions. The frequencies of each class of ATAC-seq peaks are indicated. (**E–G**) TET-regulated regions are enriched for E2A and PU.1 binding sites. (**E**) Genome browser view of the 3′ end at the Igκ locus. From top to bottom are tracks for ATAC-seq and 5hmC/CMS-IP tracks of WT or *Tet2/3* DKO pro-B cells (combined from two replicates), followed by E2A, PU.1, and H3K4me1 tracks from WT pro-B cells. Rectangles show the locations of the intronic (iEκ), 3′ (3′Eκ) and distal Eκ (dEκ) enhancers. (**F**) Left panel shows the ATAC-seq signal from WT (1$^{st}$ column) and DKO (second column) across indicated type of regions, with each horizontal line representing one region/locus. Note that the commonly accessible and randomly sampled regions are compressed compared to WT and DKO DARs in order to accommodate all regions. Right panel shows PU.1 and E2A ChIP-seq signal from published datasets across all regions. Randomly sampled regions are included for comparison. (**G**) Mean RPKM signals of PU.1 (*left*) and E2A (*right*) were shown as a histogram. Note that the WT DARs (blue) have the stronger enrichment of both transcription factors compared to other regions. Analysis of additional transcription factors can be found in *Figure 4—figure supplement 2*.

The following figure supplements are available for figure 4:

**Figure supplement 1.** Identification of sequence motifs enriched in differential accessible regions.
**Figure supplement 2.** Binding pattern of additional B-cell-specific and general transcription factors at accessible regions.
**Figure supplement 3.** Tet2 and Tet3 regulate the accessibility and demethylation of enhancers associated with genes critical for B cell development.

transcription factors in pro-B cells (*Heinz et al., 2010*; *Lin et al., 2010*) either did not show a consistent relation to accessibility (EBF, Oct2) or were similar between WT>DKO DARs and other accessible regions (Foxo1; *Figure 4—figure supplement 2*).

Other examples of decreased accessibility in *Tet2/3* DKO compared to WT pro-B cells are shown for the *CD79α* and *Foxo1* loci; some of these sites coincide with E2A and/or PU.1 binding sites identified by ChIP-seq (*Heinz et al., 2010*; *Lin et al., 2010*), and CpG sites with low methylation (5mC + 5 hmC) at these regions in WT cells show increased methylation in *Tet2/3* DKO cells (*Figure 4—figure supplement 3*). Together these data point to a significant association between chromatin accessibility, TET protein activity (i.e. presence of 5hmC) and the binding of two transcription factors, PU.1 and E2A, with key roles in B cell development.

Notably, consistent with a previous study (*Benner et al., 2015*), a fraction of the commonly accessible regions (especially those with the lowest ATAC-seq signals) corresponded to regions with significant CTCF binding in pro-B cells (*Lin et al., 2010*), suggesting these regions are potent insulators (*Song et al., 2011*). In contrast, CTCF binding sites were not enriched in the WT>DKO and DKO>WT DARs (*Figure 4—figure supplement 2*). These results suggest that regulation of insulator accessibility is likely TET-independent and imply that multiple mechanisms exist to regulate accessibility in different chromatin contexts.

## Increased DNA methylation at enhancers correlates with decreased accessibility

In addition to mapping 5hmC by CMS-IP, we characterized the methylome of WT and *Tet2/3* DKO pro-B cells by whole-genome bisulfite sequencing (WGBS), which does not distinguish 5hmC from 5mC, or 5fC and 5caC from unmodified C (*Huang et al., 2010*; *Pastor et al., 2013*). Based on the striking global loss of 5hmC in DKO pro-B cells as judged by the highly selective CMS-IP method (*Huang et al., 2012*; *Pastor et al., 2012*)(*Figure 3—figure supplement 2B*), all unconverted C's identified by WGBS were assigned to 5mC. Notably, the methylation landscapes were similar

between WT and *Tet2/3* DKO pro-B cells, with 872 regions showing increased methylation and 258 showing decreased methylation in DKO pro-B cells (*Figure 5*). Analysis of these differentially methylated regions (DMRs) showed that DMRs with increased methylation in DKO cells compared to WT (DKO>WT DMRs) were concurrently less accessible (*Figure 5A*, left; *Figure 5—figure supplement 1A*, compare first and second columns), and had originally been modified with 5hmC in WT pro-B cells (*Figure 5A*, *right*; *Figure 5—figure supplement 1A*, see third column), suggesting that TET activity is required for antagonizing DNA methylation and maintaining accessibility at these regions. Consistent with the role of TET in regulating enhancer accessibility (shown above), the majority of regions with increased methylation in DKO cells compared to WT (72.5%) show characteristics of enhancers: location distal to transcription start sites (*Figure 5B*) and relative enrichment for H3K4me1 over H3K4me3 (*Figure 5C*) and were enriched for E2A and PU.1 binding (*Figure 5—figure supplement 1A*; compared to randomly sampled regions in *Figure 5—figure supplement 1B*). Together these results emphasize the strong correlation between chromatin accessibility, TET enzymatic activity based on generation of 5hmC, and maintenance of DNA demethylation at the enhancer regions marked by 5hmC. It has yet to be established, however, whether loss of 5hmC or

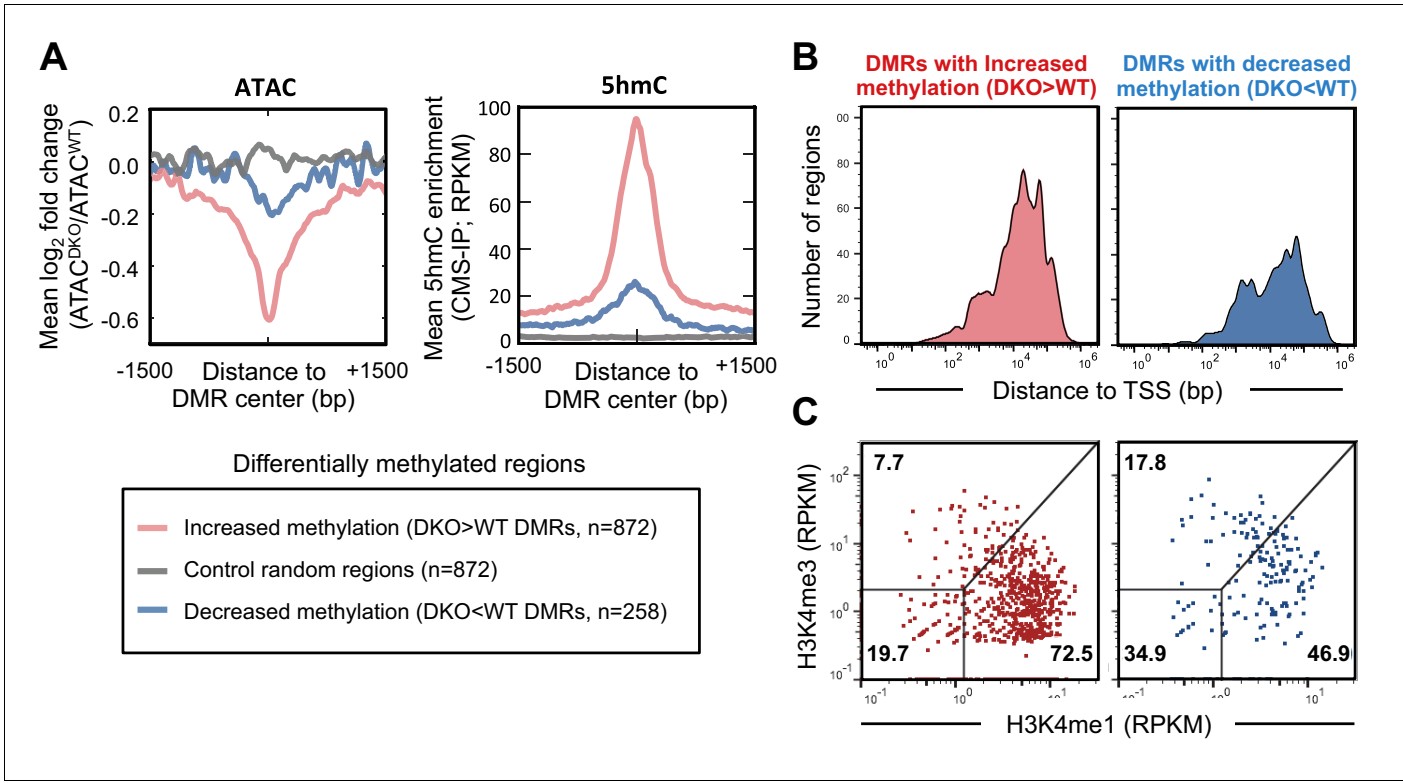

**Figure 5.** TET proteins demethylate and modulate the accessibility of enhancers. DNA isolated from WT and *Tet2/3* DKO pro-B cells (two replicates each) were bisulfite-treated and sequenced to analyze the genome-wide DNA cytosine methylation. Differential methylation regions were analyzed as described in Materials and Methods, and 872 and 258 regions were found to have more methylation in DKO ('DKO > WT DMRs' in red) and WT ('DKO<WT DMRs' in blue), respectively. (A) Regions with increased methylation in *Tet2/3* DKO are highly enriched in 5hmC and more accessible in WT compared to *Tet2/3* DKO. *Left*, the log2 ratio between ATAC^WT and ATAC^DKO was calculated for the indicated regions (bin size = 10 bp) and the means were plotted. *Right*, 'DKO > WT methylation' regions were marked with 5hmC in WT pro-B cells. The mean RPKM values for 5hmC enrichment (detected by CMS-IP) were plotted for each set of regions. (B–C) Differentially methylated regions bear the features of enhancers. (B) Distance of differential methylated sites to the closest transcription start sites (TSSs) was plotted as histogram as in *Figure 4C*. (C) Relative enrichment of H3K4me1 and H3K4me3 at the differential methylated regions was plotted as in *Figure 4D*.

The following figure supplement is available for figure 5:

**Figure supplement 1.** TET proteins demethylate and modulate the accessibility.

increased 5mC plays a more significant role in the decreased chromatin accessibility in *Tet2/3* DKO pro-B cells.

## Functional and physical interaction between TET proteins and E2A/PU.1

Our results so far pointed to a strong functional interaction between TET proteins and two key transcription factors expressed in pro-B cells, E2A and PU.1: briefly, we observed TET activity (deposition of 5hmC and subsequent demethylation) at enhancers marked by the binding of E2A and PU.1, but not EBF, Foxo1 or Oct2 (*Figures 4* and *5*). We confirmed the functional interaction by transducing BCR-Abl1-transformed pre-B cells with lentiviruses encoding two independent shRNAs against E2A and PU.1 respectively, then examining the DNA modification status of the 3′ and distal Eκ enhancers by bisulfite sequencing (*Figure 6*). The shRNAs reduced the amount of E2A and PU.1 proteins substantially (*Figure 6—figure supplement 1A and B*), and increased the average level of DNA methylation at the first and second CpGs of the distal Eκ enhancer by 2- to 3-fold (*Figure 6A and B*), a magnitude similar to that observed in *Tet2/3* DKO pro-B cells (*Figure 3D*).

Based on these data, we tested the possibility of a physical interaction between TET proteins and E2A and/or PU.1. We found that indeed, TET2 co-immunoprecipitated with E2A and PU.1 (*Figure 6C*); the interaction was direct, since we used benzonase (a nuclease that degrades both DNA and RNA) and the DNA intercalator ethidium bromide to prevent false positive interactions indirectly mediated by contaminating nucleic acids in cell extracts. These findings suggested that E2A and PU.1 recruit TET proteins to enhancers, where they increase chromatin accessibility by depositing 5hmC and facilitating DNA demethylation.

We tested the recruitment hypothesis by using ChIP-qPCR to evaluate the presence of TET2 at the 3′ and distal Eκ enhancers, in pro-B cells depleted of E2A or PU.1 (*Figure 6A and B*). Indeed, recruitment of TET proteins to the distal Eκ enhancer was significantly diminished in cells depleted for either E2A or PU.1 (*Figure 6D and E*). Notably, recruitment of TET2 to the 3′Eκ enhancer was not significantly affected (*Figure 6D and E*); this is entirely consistent with the fact that we observed increased methylation under the same conditions at the distal Eκ enhancer but no increase, or a much smaller increase, at the 3′Eκ enhancer (*Figure 6A and B*). The difference could be due to a requirement for additional factors (other than E2A or PU.1 alone) at the 3′Eκ enhancer, or to a threshold effect stemming from the stronger binding of E2A and PU.1 to the 3′Eκ compared to the dEκ enhancer (*Figure 4E*).

PU.1 has been proposed to be a 'pioneer' transcription factor capable of association with 'closed' chromatin (*Ghisletti et al., 2010*; *Heinz et al., 2010*). Consistent with this notion, we found that the genome-wide binding of PU.1 was virtually indistinguishable between WT and *Tet2/3* DKO pro-B cells (*Figure 6—figure supplement 2*), suggesting that PU.1 is upstream of TET and enlists TET activity to modulate the accessibility of bound enhancers. Nonetheless, these data suggest a tight functional relationship between TET proteins and the key transcription factors during B cell development.

## Tet2 enzymatic activity is necessary to promote Igκ expression and chromatin accessibility

The availability of the in vitro system allowed us to ask whether TET catalytic activity was required for Igκ expression (*Figure 7*). For unknown reasons, we could not efficiently express full-length Tet2 or Tet3 in pro-B cells, but we did achieve short-lived expression of isolated Tet2 catalytic domain (Tet2CD) when we used a retroviral vector containing a blasticidin resistance gene (*Figure 7A*; the cells expressed Tet2CD for ~1 week after drug selection was initiated, after which expression declined rapidly). Under these conditions, *Tet2/3* DKO BCR-Abl pre-B cells reconstituted with catalytically-active Tet2CD showed increased Igκ germline and IgCκ transcription (*Figure 7B*); they also displayed decreased DNA methylation at the 3′ and distal Eκ enhancers, compared to mock-transduced cells (*Figure 7C*). Cells transduced with a catalytically-inactive version of Tet2CD (HxD mutant (*Ko et al., 2010*; *Tahiliani et al., 2009*) were functionally inert (*Figure 7B and C*), despite the fact that the mutant protein was recruited to the 3′ Eκ and distal Eκ enhancers as or almost as effectively as the catalytically active version (*Figure 7D*). Expression of Tet2CD, but not its HxD mutant, also restored chromatin accessibility at the 3′Eκ and distal Eκ enhancers (*Figure 7E*) as well as at a genome-wide level (*Figure 7F*) in *Tet2/3* DKO pro-B cells. Importantly, re-expression of Tet2CD was

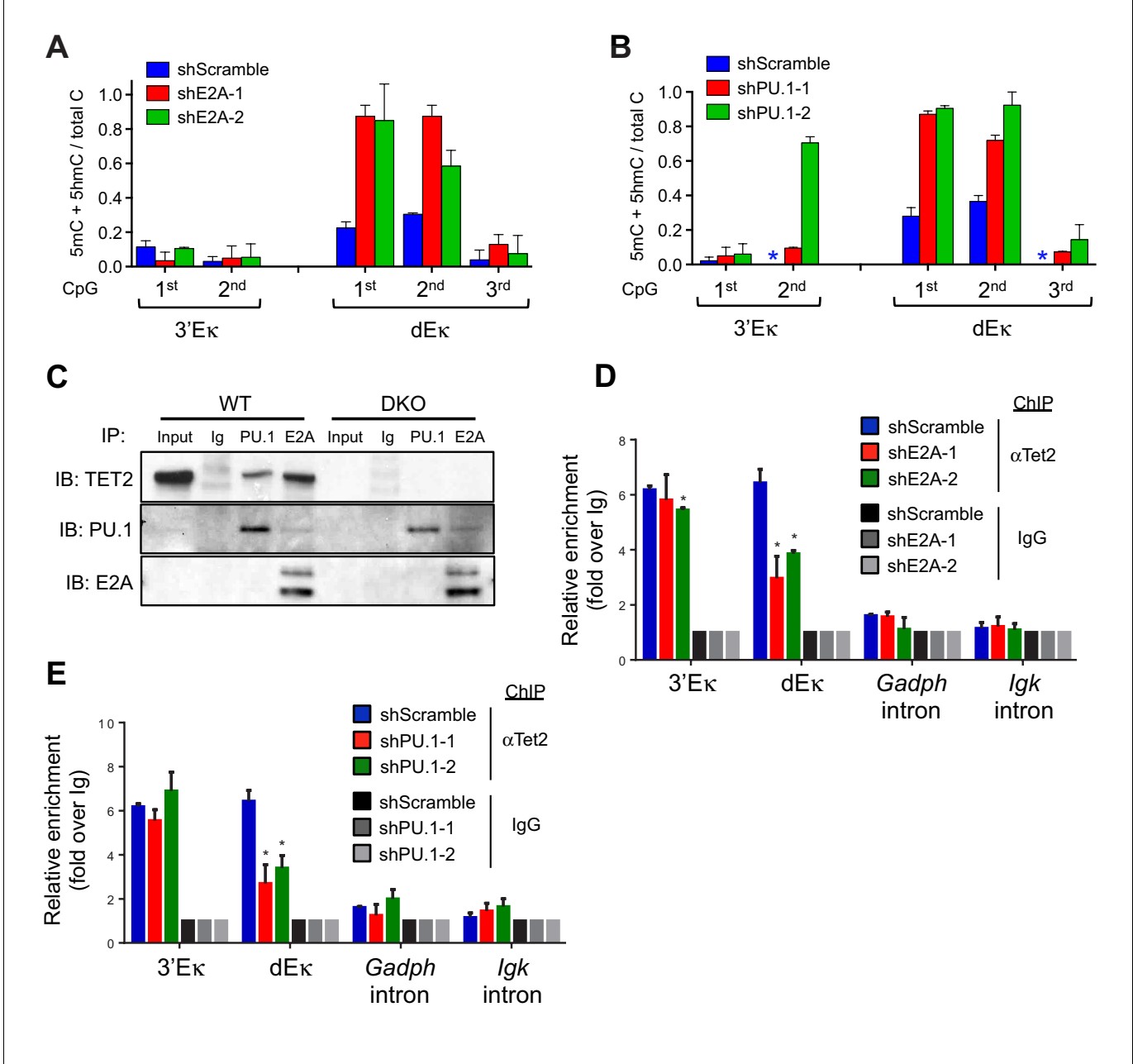

**Figure 6.** Lineage-specific transcription factors cooperate with TET protein in inducing Igκ enhancer demethylation. (**A**) Depletion of E2A increases DNA modification (5mC + 5 hmC) at the distal Eκ enhancer. WT BCR-Abl-transformed pre-B cells were transduced with two independent shRNAs against E2A or scrambled shRNA as a control. Methylation of 3′Eκ and dEκ was determined by bisulfite sequencing in the control or E2A-depleted cells. E2A knockdown is verified by immunoblotting (**Figure 6—figure supplement 1A**). Data are the summary of two independent experiments. Error bars show the range of duplicates. (**B**) Depletion of PU.1 increases DNA modification (5mC + 5 hmC) at the distal Eκ enhancer. As in (**A**), methylation of 3′Eκ and dEκ enhancers was analyzed by bisulfite sequencing after PU.1 knockdown. PU.1 knockdown is verified by immunoblotting (**Figure 6—figure supplement 1B**). Error bars show the range of duplicates. (**C**) Tet2 directly interacts with PU.1 and E2A. E2A and PU.1 were immunoprecipitated from WT or *Tet2/3* DKO BCR-Abl pre-B cell nuclear extract in the presence of ethidium bromide and benzonase to prevent indirect 'interaction' via DNA. Co-immunoprecipitated proteins were probed with anti-Tet2. Input loaded was 2.5%. Note that a different secondary antibody was used for PU.1 and E2A to avoid the interference of IgH and IgL and thus weaker signal. (**D–E**) E2A and PU.1 facilitate the binding of Tet2 to dEk. E2A (**D**) and PU.1 (**E**) were depleted by shRNAs as in (**A**) and (**B**) and the association of Tet2 to Igκ enhancers or control regions (*Gadph* and *Igκ* introns) were assessed by ChIP-qPCR. Data are representative for at least two experiments.

The following figure supplements are available for figure 6:

*Figure 6 continued on next page*

*Figure 6 continued*

**Figure supplement 1.** Efficient knockdown of E2A and PU.1.
**Figure supplement 2.** Tet2/3-deficiency has limited effect on genome-wide PU.1 binding.

able to restore the VκJ rearrangement in *Tet2/3* DKO pre-B cells (*Figure 7G*), demonstrating the functional relevance of enhancer accessibility to biological outcome.

## TET proteins regulate the expression of IRF4

In addition to decreased levels of Igκ germline transcripts, *Tet2/3* DKO cells showed a significant decrease in *Irf4* mRNA and protein levels (*Figure 3D*, *Figure 3—figure supplement 2C*). Expression of IRF4 protein and mRNA required TET catalytic activity, since re-expression of Tet2CD restored mRNA and protein levels of IRF4 whereas the Tet2CDHxD mutant did not (*Figures 7B* and *8A*). Thus in addition to VκJ rearrangement, TET proteins regulate the expression of IRF4, which binds to 3'Eκ and potentially dEκ and regulates Ig light chain rearrangement and control pre-B cell development (*Ma et al., 2006*). Nonetheless, over-expression of IRF4 marginally induced Igκ gene transcription and did not affect 3'Eκ and dEκ methylation in the absence of TET (*Figure 8B and C*). These results suggested that IRF4 is induced via TET catalytic activity and then functions downstream of TET proteins in regulating early B cell development.

## Discussion

We have shown that mice with a germline deletion of *Tet2* in conjunction with *Mb1Cre*-driven deletion of *Tet3* display a block in B cell development at the pro-B to pre-B transition, reflective of decreased Igκ rearrangement and expression. A profound loss of TET function is necessary, since mice lacking only Tet2 or Tet3 in B cells have no obvious B cell phenotypes. By acute Cre-mediated deletion of *Tet3* in cultured pro-B cells and rescue with Tet2, we have demonstrated that TET proteins concurrently regulate Igκ germline transcription and the DNA methylation status of the 3' and distal Eκ enhancers, through a mechanism that requires TET catalytic activity.

There are well-established correlations among histone/ DNA modifications, chromatin accessibility and B cell development (*Benner et al., 2015*; *Thurman et al., 2012*). DNA cytosine demethylation, most likely mediated by TET enzymes, has a role in organizing genome domains by affecting the binding of CTCF (*Benner et al., 2015*; *Flavahan et al., 2016*). Moreover, conditional *Dnmt3a/b* deletion results in early Vκ-Jκ rearrangement, increased expression of Igκ germline transcripts, and decreased DNA methylation at Igκ enhancers (*Manoharan et al., 2015*). BRG1, the catalytic (ATPase) component of the SWI/SNF chromatin remodeling complex, and the chromatin reader BRWD1, have both been implicated in early B cell development (*Bossen et al., 2015*; *Mandal et al., 2015*). Indeed, Yancopoulos and Alt postulated more than 30 years ago that Ig locus accessibility is critical for Ig germline transcription and V(D)J rearrangement (*Yancopoulos and Alt, 1985*). Our data suggest that Tet2 and Tet3 maintain chromatin accessibility at Igκ locus enhancers as well as at many other promoters and enhancers in pro-B cells. While many protein complexes can modulate chromatin accessibility, the fact that acute deletion of TET function almost eliminates Igκ germline transcription, in a manner that can be rescued by a TET catalytic domain, implies that the TET enzymatic activity has a primary and essential role.

The expected consequence of decreased accessibility is decreased binding of transcription factors to the poorly accessible sites. We have shown that genomic regions that are less accessible in *Tet2/3* DKO compared to WT pro-B cells are enriched for E2A and PU.1 binding motifs, and conversely, a subset of validated E2A and PU.1 binding sites, including the Igκ enhancers, are less accessible in *Tet2/3* DKO pro-B cells. Notably, both transcription factors are involved in maintaining the demethylated status of the distal Eκ enhancer. PU.1, a key transcription factor in both the myeloid and lymphoid lineages, functions in macrophages as a 'pioneer' transcription factor capable of enhancing chromatin accessibility by binding linker regions between nucleosomes, thereby facilitating the modification of neighboring histones and the binding of other transcription factors (*Ghisletti et al., 2010*; *Heinz et al., 2010*). Consistent with a previous finding in macrophages that

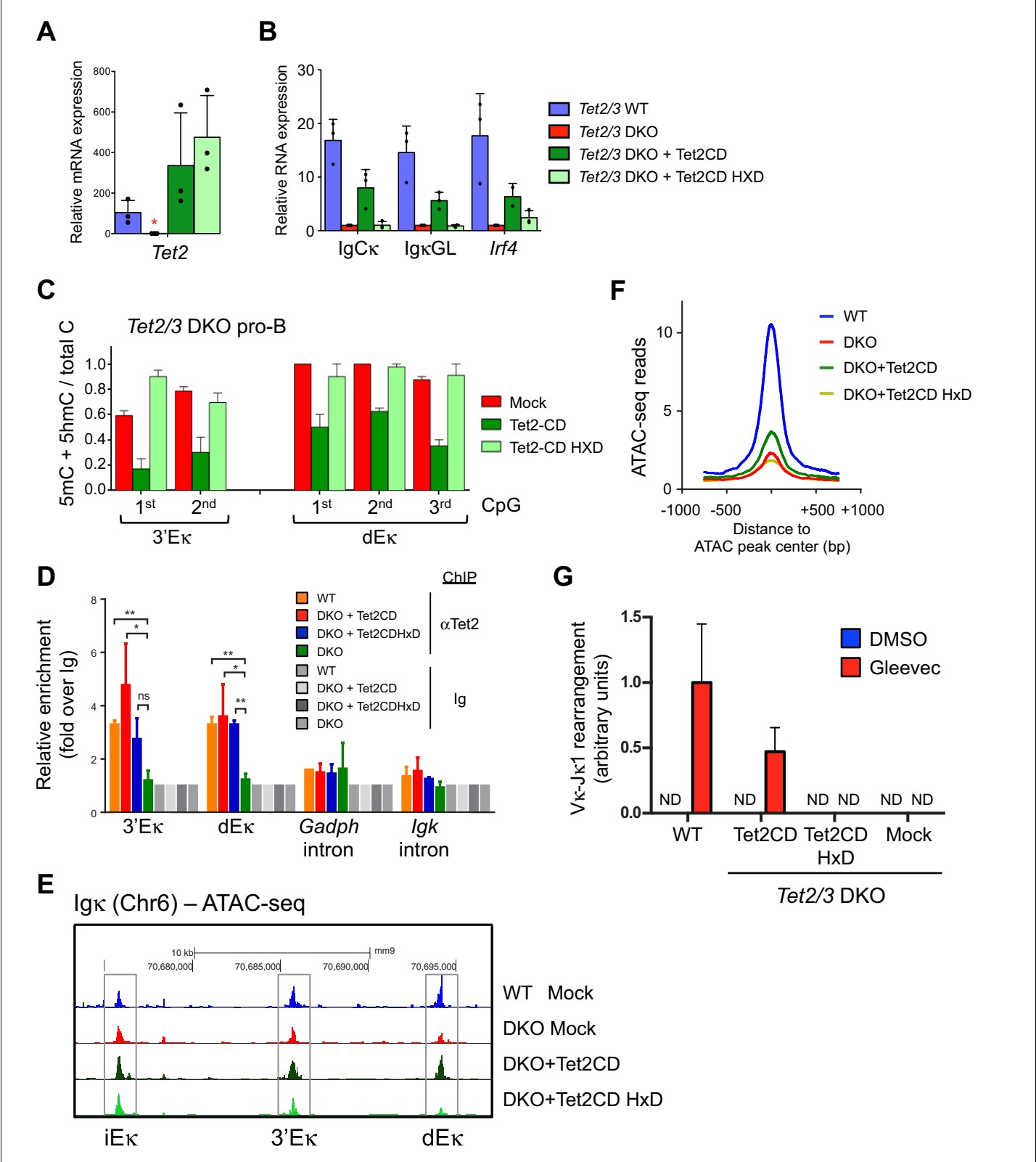

**Figure 7.** Tet2CD rescues Igκ expression, promotes demethylation and accessibility of 3'Eκ and distal Eκ enhancers. (A–B) The enzymatic activity of Tet2 is required to promote Irf4 expression and Igκ germline transcription. BCR-Abl-transformed *Tet2/3 DKO* pre-B cells were transduced with retrovirus expressing Tet2CD, the corresponding Tet2CD HxD catalytic inactive mutant or empty vector (mock) together with a blasticidin resistance gene. Seven days post blasticidin selection, expression of *Tet2* (A), Cκ, Igκ germline transcripts and *Irf4* was determined by real time PCR (B). Error bars

*Figure 7 continued on next page*

*Figure 7 continued*

represent the standard deviation of three independent experiments. *, p<0.01 in Student's *t* test. (**C**) The enzymatic activity of Tet2 is required to promote 'demethylation' (loss of 5mC + 5 hmC) at Igκ enhancers. BCR-Abl-transformed *Tet2/3 DKO* pre-B cells were transduced as in (**B**), and the DNA modification status of the 3' and distal Eκ enhancers was analyzed by bisulfite sequencing. Error bars indicate the range of values obtained in two independent experiments. (**D**) Tet2 associates with the Igk enhancers. WT, *Tet2/3* DKO, and DKO transduced with Tet2CD or Tet2CD HxD Abl-transformed pre-B cells were used as input and Tet2 binding to indicated regions was detected by ChIP-qPCR. The signal was normalized to corresponding samples immunoprecipitated with Ig. (**E**) Tet2CD restores chromatin accessibility at the Igκ enhancers in *Tet2/3 DKO* cells. *Tet2/3 DKO* pro-B cells were transduced with retrovirus containing Tet2CD-IRES-Thy1.1 (Tet2CD), Tet2CD HxD mutant-IRES-Thy1.1 (Tet2CD HxD) or empty vector (mock). Thy1.1$^+$ cells were sorted and chromatin accessibility was analyzed through ATAC-seq. (**F**) Tet2 requires its enzymatic activity in promoting chromatin accessibility. TET-regulated accessible regions (WT>DKO DARs) were identified as in *Figure 4A*, and chromatin accessibility of these regions were plotted in WT pro-B cells, or *Tet2/3 DKO* pro-B cells reconstituted with Tet2CD, Tet2CD HxD or empty vector (mock). Histogram shown is distribution of ATAC-seq reads (normalized to 10 million reads depth) over 750 bp downstream or upstream of the peak center. Verification of Tet2 expression by immunoblotting is shown in *Figure 8A*. (**G**) Tet2CD restores VκJ recombination in Tet2/3 DKO pre-B cells. Abl-transformed pre-B cells transduced with Tet2CD or Tet2CD HxD were treated with Gleevec to induce VκJ rearrangement and analyzed as in *Figure 2C* by ligation-mediated-PCR.

both TET2 and DNMT3b co-immunoprecipitate with PU.1 (*de la Rica et al., 2013*), we report here that TET2 co-immunoprecipitates with both PU.1 and E2A in developing B cells. Loss of TET2 and TET3 has a limited effect on the binding of PU.1 (*Figure 6—figure supplement 2*), suggesting that PU.1, and potentially E2A, recruit TET proteins to Igκ and other enhancers where they render the enhancers permissive for additional and/or more stable binding of those and other transcription factors (*Figure 8D*).

IRF4 and IRF8 are functionally redundant for the rearrangement of Ig light chains during pro-B to pre-B transition (*Lu et al., 2003*), while EBF1 is imperative for the pre-pro-B to pro-B transition (*Lin and Grosschedl, 1995*). Our data show that in addition to regulating Igκ germline transcription, TET2 and TET3 proteins are required for the expression of IRF4 and, to a lesser extent, IRF8. Ectopic IRF4 expression induced Igκ germline transcription as well as premature Igκ rearrangement in WT pro-B cells (*Bevington and Boyes, 2013*), but ectopic expression of IRF4 in *Tet2/3* DKO cells only marginally increased Igκ germline transcription. Moreover, expression of IRF4 in *Tet2/3* DKO cells had no effect on the DNA modification status of the Igκ enhancers, suggesting that IRF4 and IRF8 function at a late stage, after transcription factors such as PU.1 and E2A have cooperated with TET enzymes to facilitate enhancer accessibility. Interestingly, CpG dinucleotides are present in at least one potential IRF4 binding site in both the 3'Eκ and dEκ enhancers (*Figure 3—figure supplement 1A*), hinting that TET-mediated demethylation may be required for efficient binding of IRF4 at these enhancers. Finally, although we did not examine EBF1 because of lack of enrichment of its binding sites in TET-regulated accessible regions of the genome, EBF1 has been shown to increase both chromatin accessibility and DNA demethylation in B progenitor cells (*Boller et al., 2016*), and to interact with TET2 in a cancer cell line (*Guilhamon et al., 2013*).

Our reconstitution experiments show clearly that TET-mediated changes in DNA modification status and chromatin accessibility can be rescued by re-expression of the TET2 catalytic domain in *Tet2/3* DKO pro-B cells. Although for technical reasons, we were unable to reconstitute the cells with full-length Tet2, it is not surprising that we observed a substantial restoration with the catalytic domain alone: many documented interactions of TET proteins with their partners involve the catalytic domain (e.g. TET1-OGT (*Balasubramani and Rao, 2013*), TET2-IDAX (*Ko et al., 2013*)), and Tet2 lacks a covalently associated CXXC domain that would be expected to target it to genomic regions containing unmethylated CpGs (*Ko et al., 2013*).

Our current study focuses on the function of Tet2 and Tet3 in early B cell development, but our data indicate that TET proteins also have a role in mature B cells. Specifically, we observed an age-related decline of mature B cells in *Tet2/3* DKO mice, suggesting that Tet2 and Tet3 are important for B cell homeostasis. Despite this, we observed a fully penetrant development of B cell lymphomas in all *Tet2/3* DKO mice, consistent with our recent finding that acute deletion of Tet2 and Tet3 in hematopoietic stem/ precursor cells results in acute myeloid leukemia (*An et al., 2015*). A recent study characterizing DNA methylation dynamics during human B cell differentiation revealed distinct demethylation patterns between early B cells and mature B cells (*Kulis et al., 2015*); demethylation

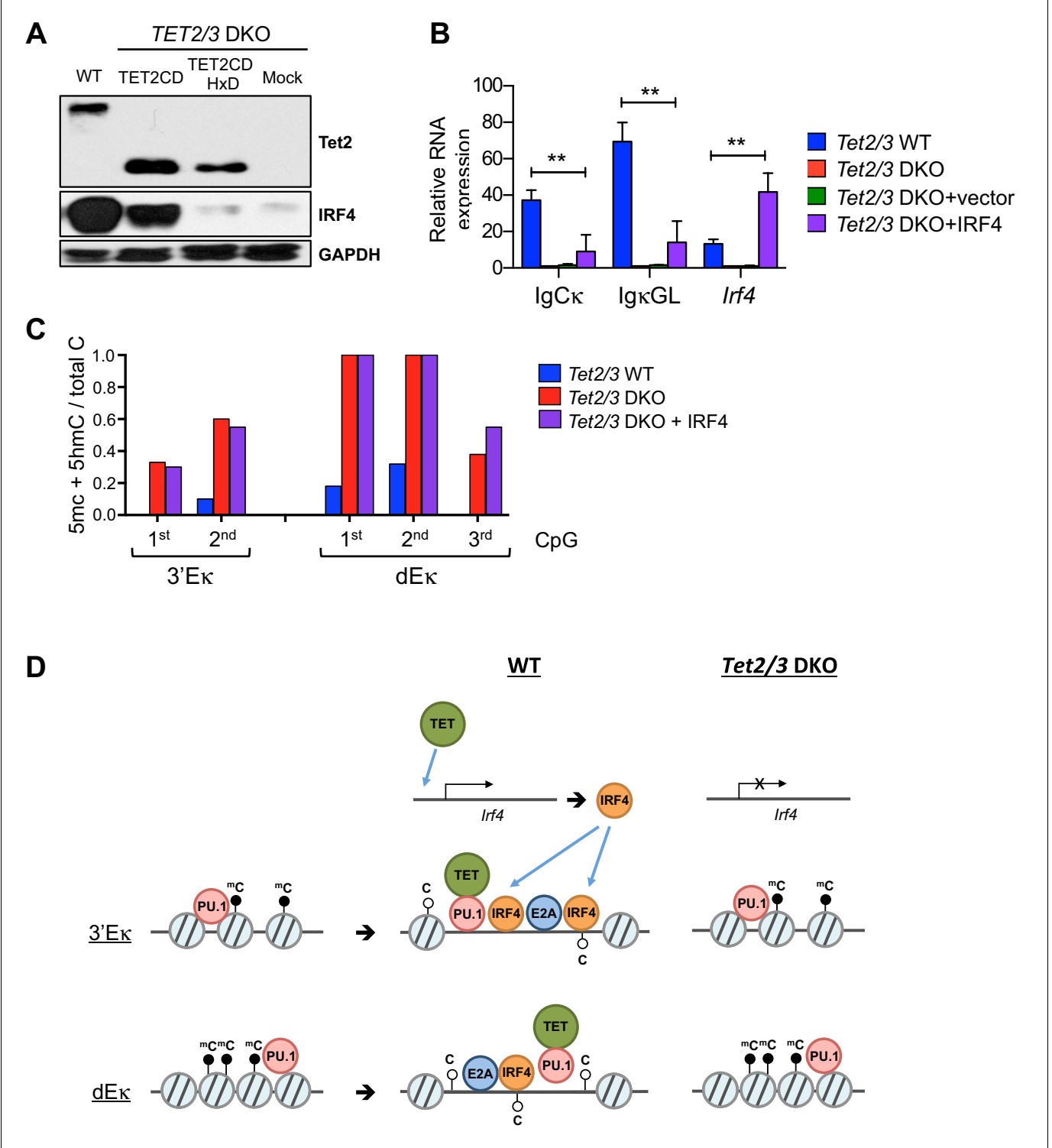

**Figure 8.** TET regulates expression of IRF4. (**A**) Immunoblotting for Tet2 and IRF4 in Tet2/3 DKO BCR-Abl cells transduced with Tet2CD or Tet2CD HxD mutant. (**B**) Reconstitution with Irf4 only marginally rescued Igκ expression in the *Tet2/3 DKO* cells. Tet2/3 DKO cells were transduced with retrovirus expressing IRF4 and IgCκ and IgκGL transcription was analyzed by qRT-PCR. (**C**) Re-expression of IRF4 has no effect on enhancer methylation in the absence of Tet2/3. Enhancer methylation status of cells form (**B**) was analyzed by bisulfite sequencing. (**D**) Working model of TET-mediated regulation of the Igκ locus. During early B cell development, the pioneer transcription factor PU.1 binds at various locations including the Igκ enhancers (e.g. 3'Eκ and dEκ shown here; *left*). Subsequently, presumably after IgH rearrangement, PU.1 and potentially E2A recruit TET proteins, facilitating the deposition of 5hmC and DNA demethylation. These TET-dependent activities increase enhancer accessibility, likely resulting in increased binding of additional

*Figure 8 continued on next page*

*Figure 8 continued*

transcription factors (*middle*). In the absence of TET proteins, the enhancers remain methylated and are less accessible for transcription factors (*right*). In addition, TET proteins regulate the expression of IRF4, a transcription factor important for the induction of Igκ rearrangement (*top*).

in early human B cell development mainly occurs at enhancers, consistent with our observations in mice.

Bestor (**Bestor et al., 2015**) and Schübeler (**Schübeler, 2015**) have persuasively argued that DNA methylation and demethylation are not 'instructive' for gene transcription but rather are byproducts of altered transcription. In contrast, our data suggest that at least at Igκ enhancers in pro-B cells, the increase in DNA methylation that results from TET loss-of-function can be a determining ('instructive') factor for the observed decrease in Igκ germline transcription. However, our data also show that acute depletion of E2A results in increased DNA methylation at the distal Eκ enhancer, thus connecting transcription factor binding to DNA demethylation as postulated (**Bestor et al., 2015**; **Schübeler, 2015**). The simplest scenario is a positive feedback loop in which E2A, PU.1 and other transcription factors (**de la Rica et al., 2013**; **Guilhamon et al., 2013**; **Wang et al., 2015**) recruit TET proteins to enhancers, thus facilitating TET-mediated DNA modification; the recruited TET proteins, via DNA demethylation, maintain chromatin accessibility at these regions, promoting increased binding of E2A and other transcription factors such as PU.1 (**Figure 8D**). Future studies will reveal how lineage-specific transcription factors, chromatin regulatory proteins and TET enzymes together orchestrate cell fate determination and gene expression in B cells and other cell types.

## Materials and methods

### Mice

$Tet2^{fl/fl}$, $Tet2^{-/-}$ and $Tet3^{fl/fl}$ mice were generated as described (**Ko et al., 2011, 2010**, **2015**). *Mb1Cre* mice were obtained from Dr. Changchun Xiao and originally from Dr. Michael Reth (**Hobeika et al., 2006**). All mice were in the C57BL/6 background.

### RNA extraction, cDNA synthesis and quantitative Real-Time PCR

Total RNA was isolated with RNeasy plus kit (Qiagen, Germnay) and cDNA was synthesized using SuperScript III reverse transcriptase (Thermo Fisher, Waltham, MA). Quantitative PCR was performed using FastStart Universal SYBR Green Master mix (Roche, Germany) on a StepOnePlus real-time PCR system (Thermo Fisher). Gene expression was normalized to *Actb*. Primers were listed in *Supplementary file 1*.

### Derivation of *Tet2/3* DKO pro-B cells in vitro

Total bone marrow cells from $Tet2^{-/-}Tet3^{fl/fl}$ mice were cultured in α-MEM medium with 20% FBS and IL-7 (Peprotech, Rocky Hill, NJ; 10 ng/ml) on OP9 feeder cells for seven days, and transduced with retrovirus containing Cre-IRES-GFP. GFP⁺ cells were sorted 5–7 days after transduction and maintained as above. Cells were routinely monitored for mycoplasma contamination and were negative.

To obtain BCR-Abl transformed pre-B cells, pro-B cells were transduced with retrovirus encoding a BCR-ABL fusion protein (p210, Addgene). Three days after transduction, cells were removed from IL-7 and OP9. To induce Igκ locus rearrangement, cells were treated with Gleevec (10 μM) for indicated times, and double strand breaks at the Jκ1 RSS1 region were quantified by ligation-mediated PCR as described (**Curry et al., 2005**).

To knockdown E2A and PU.1 in WT BCR-ABL transformed pre-B cells, cells were transduced overnight with lentivirus generated from pLKO.1 based shRNAs, including shScramble, shE2A-1, shE2A-2, shPU.1–1, and shPU.1–2 similar to previously described (**Martinez et al., 2015**). Cells were selected with puromycin at 18 hr post-transduction and continuously for 72 hr. shRNA target sequences are as follow: shE2A-1, CTGCACCTCAAGTCGGATAAG; shE2A-2, TTTGACCCTAGCCG-GACATAC; shPU.1–1, AACAGAGCTGAACAGTTTGGG; shPU.1–2,TTCTGATACGTCATGCGCTTG.

## Immunoprecipitation

Nuclear extracts from WT and *Tet2/3* DKO Abl-preB cells were immunoprecipitated with anti-PU.1 and anti-E2A in the presence of benzonase (Sigma, St. Louis, MO; final 500 U/mL) and ethithum bromide (Sigma; 100 µg/mL) to prevent indirect pulldown via DNA binding. Resulting proteins and 2.5% of input nuclear lysate were blotted with anti-TET2, -PU.1, and -E2A antibodies. HRP-conjugated polyclonal goat-anti-rabbit antibodies were used for detecting TET2. For PU.1 and E2A detection, blots were stripped, and probed with primary and then secondary monoclonal antibody specific for rabbit IgG conformation (Cell Signaling Technology, Danvers, MA) to minimize reactivity with IgH and IgL at around 50 and 20 kDa, respectively. Note that the overall signal was lower for E2A and PU.1 due to the secondary antibody used.

## Flow cytometry and antibodies

For cell surface staining, cells were stained with indicated antibodies in FACS buffer (1% FBS and 0.1% NaN$_3$ in PBS) for 30 mins on ice, washed, and fixed in 1% paraformaldehyde in PBS. Cytofix/Cytoperm kit (BD Bioscience, Franklin Lakes, NJ) was used for intracellular staining. Samples were analyzed with LSR Fortessa (BD Biosciences) and FlowJo (FlowJo LLC, Ashland, OR). The following antibodies were used in this study and were purchased from BD, eBioscience (San Diego, CA), or Biolegend (San Diego, CA): CD19 (clone 6D5), B220 (clone RA-3-6B2), CD25 (PC61), IgM (R6-60), IgD (11–26c.2a), CD43 (S7), CD179α (R3), cKit (2B8), Igκ (187.1), Igμ (Il/41). The following antibodies were used for immunoblotting and ChIP: anti-E2A (Santa Cruz, Dallas, TX; V18), anti-PU.1 (Santa Cruz, T-21), anti-IRF4 (Santa Cruz, M-17) and anti-TET2 (Abcam, United Kingdom;ab124297).

## Igκ rearrangement analysis

To analyze *Igκ* rearrangements, a degenerate Vκ primer (GGC TGC AGS TTC AGT GGC AGT GGR TCW GGR AC) and an Igκ intron primer (AAC ACT GGA TAA AGC AGT TTA TGC CCT TTC) were used; Quantitation of Vκ-Jκ1 rearrangement was done by qPCR with these primers: degVκ: GGC TGC AGS TTC AGT GGC AGT GGR TCW GGR AC and κ-J1-R: AGC ATG GTC TGA GCA CCG AGT AAA GG.

## Enrichment-based 5hmC-mapping (CMS-IP)

CMS-IP was performed similar to pervious described (*Huang et al., 2012*; *Pastor et al., 2011*). Briefly, genomic DNA from in vitro cultured pro-B cells was isolated with PureLink Genomic DNA kit (Thermo Fisher) and were spiked-in with *c*l857 *Sam7* λDNA (Promega,Madison, WI) and PCR-generated, hmC-containing puromycin-resistant gene at a ratio of 200:1 and 100,000:1, respectively. DNA was sheared with a Covaris E220 (Covaris), end-repaired, A-tailed, ligated with methylated Illumina adaptors (NEB, Ipswich, MA), and bisulfite-converted (MethylCode Bisulfite Conversion Kit, Thermo Fisher). Bisulfite-converted DNA was denatured and immunoprecipitated with anti-CMS serum. Immunoprecipitated DNA was PCR-amplified with barcoded primers (NEBNext Multiplex Oligos for Illumina, NEB) for 15 cycles with Kapa HiFi Uracil+ (Kapa Biosystems, Wilmington, MA). Resulting libraries were sequenced with a HiSeq2500 system for 50 bp paired-end reads (Illuminia, San Diego, CA). The sequence reads were mapped to mm9 with Bismap, and CMS-enriched genomic regions were identified using the 'findPeaks' command in HOMER with the 'histone' mode and default parameters (*Heinz et al., 2010*).

## Amplicon bisulfite sequencing

Genomic DNA was bisulfite-treated according to the manufacturer's instruction (MethylCode Bisulfite Conversion Kit, Thermo Fisher), and target sequences were amplified by PCR. The PCR products were TA-cloned and sequenced as individual clones. Alternatively, PCR products were ligated with Illumina sequencing adaptors, re-amplified and sequenced using a Miseq platform (Illuminia). For Sanger sequencing of individual clones, at least 15 sequences were obtained from each sample and aligned to the target sequences using BLAST. For bisulfite sequencing using Miseq, reads were aligned to the targets with BS-Seeker2 (*Guo et al., 2013*), and the sequencing depth was adjusted to at least 20 times for each. Average bisulfite conversion efficiencies were above 99%, calculated based on C to T conversion frequencies at non-CpG sites.

## Whole genome bisulfite sequencing

Genomic DNA from in vitro cultured WT and DKO pro-B cells was prepared similar to above for CMS-IP without spiking-in hmC-containing DNA. The adapter-ligated fragments were amplified for four cycles, cleaned up, and sequenced for an approximately 30x coverage per base. We employed BSMAP v2.74 (*Xi and Li, 2009*) to align paired-end reads from bisulfite-treated samples to the mm9 mouse reference genome allowing four mismatches. Reads mapping to multiple locations in the reference genome with the same mapping score (multiple mappers) were removed as well as the 5' ends bearing a quality lower than 20 (-v 4 w 2 r 0 -q 20). Bisulfite conversion efficiency was estimated based on cytosine methylation in non-CpG context. For all the samples the bisulfite conversion efficiency was higher than 0.9936. Duplicated reads caused by PCR amplification were removed by BSeQC v1.2.0 (*Lin et al., 2013*) applying a Poisson P-value cutoff of 1e-5. Consequently, a maximum of three stacked reads at the same genomic location were allowed and kept for further analysis. In addition, BSeQC was employed for removing DNA methylation artifacts introduced by end repair during adaptor ligation. Finally, overlapping segments of two mates of a pair were reduced to only one copy to avoid counting the same region twice during DNA methylation quantification.

## Methylation calling

To estimate CpG DNA methylation at both DNA strands, we executed the methratio.py script, from BSMAP v2.74 (*Xi and Li, 2009*) (-t 0 g 1 -x CG -i correct).

## DMR discovery

We used Regression Analysis of Differential Methylation (RADMeth; methpipe V.3.4.2) (*Song et al., 2013b*) for computing individual differentially methylated sites with the regression command, and posterior grouping into differentially methylated regions (DMRs) with the adjust and merge commands, (p.value 0.01).

## ChIP-seq

In vitro cultured pro-B cells were fixed with 1% formaldehyde at room temperature for 10 min at $1 \times 10^6$ cells/mL in media and then quench with 125 mM glycine. Nuclei were isolated and sonicated with Covaris E220 for 1200 s. Resulting chromatin was pre-cleared, immunoprecipitated with the indicated antibodies overnight, washed, and digested with proteinase K (Thermo Fisher) and RNaseA (Qiagen) at 65°C to de-crosslink overnight. DNA was purified (Zymo ChIP DNA Clean and Concentration Kit; Zymo Research, Irvine, CA) and libraries were prepared with NEBNext Ultra kit (NEB) and sequenced on Illumina Hiseq 2500 for single-end 50 bp reads.

## ATAC-seq

ATAC-seq libraries were prepared as described (*Buenrostro et al., 2013*). Briefly, 50,000 in vitro cultured pro-B cells were lysed in 25 µl ATAC-seq lysis buffer and tagged with transposase (Illumina, Nextera DNA Library Prep Kit). ATAC-seq libraries were sequenced on an Illumina Hiseq platform for paired-end $2 \times 50$ bp reads. The sequences were first mapped to the mouse genome (mm9) with bowtie (parameters: -m 1 -X 2000). The mapped reads were filtered to discard mitochondrial reads, and PCR duplicates were removed with Samtools rmdup. The accessible regions were then determined with MACS2 (-q 0.05), and the all peaks identified from two replicates from both WT and DKO were merged to generate a master peak set, which were used as regions of interest for subsequent analysis by MEDIPS (*Lienhard et al., 2014*). Differentially accessible regions were identified with FDR adjusted *p* value of 0.1. Commonly accessible regions used for analysis have adjusted *p* value higher than 0.1 and less than 4-fold difference between mean RPKM of two ATAC-seq replicates from WT and DKO. All mouse ENCODE blacklisted regions were removed (*Consortium, 2012*). The numbers for WT DARs, commonly accessible, and DKO DARs are 1303, 44668, 1192, respectively. Random genomic regions were generated with Bedtools shuffle with commonly accessible regions as input (n = 44668). HOMER annotatePeaks was used for identifying the distance from peaks to TSS. Motifs enriched in the preferential accessible regions were identified through HOMER (findMotifsGenome.pl –size given) with commonly accessible regions as background (*Heinz et al., 2010*).

To analyze the histone modifications and transcription factor binding associated with accessible regions, previous published ChIP-seq data were reanalyzed by mapping to reference genome (mm9) with Bowtie2 (*Langmead and Salzberg, 2012*). For histone marks enrichment analysis, differential and common peaks called by MEDIPS were extended for ±250 bp and the number of reads from H3K4me1 and H3K4me3 ChIP-seq datasets were retrieved and the RPKM was calculated by MEDIPS and used for scatter plot. Heatmaps and histograms were generated using Deeptools 2 (*Ramírez et al., 2016*) with a bin size of 10 bp (normalized using RPKM) and region of ±1500 bp from the peak center. Log$_2$ ratio between WT and DKO merged ATAC-seq data was calculated by Deeptools 2 bamCompare.

### Published data sets

The following published ChIP-seq data sets were used in this study: Oct2 (GSM537990), EBF (GSM546524), PU.1 (GSM537996), E2A (GSM546523), H3K4me1 (GSM546527), H3K4me3 (GSM546529), FOXO1 (GSM546525), CTCF (GSM546526), bisulfite sequencing (GSE31971).

### Ethic statement

All animal works were performed according to protocol (AP128-AR2-0516) approved by the Institutional Animal Care and Use Committee at La Jolla Institute.

## Acknowledgements

We would like to thank Dr. Cornelis Murre, Dr. Chris Benner, and Dr. Sven Heinz (UCSD) for comments and discussion; Dr. Changchun Xiao (Scripps Institute) for providing *Mb1-Cre* mice; Rao lab members for discussion and technical support; Dr. James Scott-Browne for advice on ATAC-seq analysis; Susan Togher for managing our mouse colony and for technical assistance; Cheryl Kim, Kurt van Gunst, Lara Nosworthy and Denise Hinz (LJI Flow Cytometry Core) for help with cell sorting; and Jeremy Day (LJI Functional Genomics Center) for assistance with next generation sequencing. This work was supported by NIH R01 grants AI44432 and CA151535 and LLS TRP grant 6187–12 (to AR). C-WJL was supported by a Cancer Research Institute Irvington Postdoctoral Fellowship. The authors declare no competing financial interests.

## Additional information

### Funding

| Funder | Grant reference number | Author |
| --- | --- | --- |
| Cancer Research Institute | Irvington Postdoctoral Fellowship | Chan-Wang Lio |
| Ministry of Science and Technology of the People's Republic of China | 2014CB943600 | Xing Chang |
| National Institutes of Health | AI44432 | Anjana Rao |
| National Institutes of Health | CA151535 | Anjana Rao |
| Leukemia and Lymphoma Society | 6187-12 | Anjana Rao |

The funders had no role in study design, data collection and interpretation, or the decision to submit the work for publication.

### Author contributions

C-WL, Conception and design, Acquisition of data, Analysis and interpretation of data, Drafting or revising the article, Contributed unpublished essential data or reagents; JZ, Acquisition of data, Analysis and interpretation of data; EG-A, Performed WGBS analysis and advice for general bioinformatics analysis, Analysis and interpretation of data, Drafting or revising the article; PGH, Provided critical advice, Contributed unpublished essential data or reagents; XC, Conception and design,

Acquisition of data, Analysis and interpretation of data, Drafting or revising the article; AR, Conception and design, Analysis and interpretation of data, Drafting or revising the article

## Author ORCIDs
Chan-Wang Lio, http://orcid.org/0000-0003-3876-6741

## Ethics
Animal experimentation: This study was performed in strict accordance with the recommendations in the Guide for the Care and Use of Laboratory Animals of the National Institutes of Health. All animal works were performed according to protocol (AP128-AR2-0516) approved by the Institutional Animal Care and Use Committee at La Jolla Institute.

# Additional files

## Supplementary files
• Supplementary file 1 Primer sequences.
• Supplementary file 2 Unique regions from PU.1 ChIP.

## Major datasets
The following dataset was generated:

| Author(s) | Year | Dataset title | Dataset URL | Database, license, and accessibility information |
| --- | --- | --- | --- | --- |
| Chang X, Lio CW, Zhang J, Hogan PG and Rao A | 2016 | Role of Tet proteins in B cell development | http://www.ncbi.nlm.nih.gov/bioproject/PRJNA324297 | Publicly available at the NCBI BioProject database (accession no: PRJNA324297) |

The following previously published datasets were used:

| Author(s) | Year | Dataset title | Dataset URL | Database, license, and accessibility information |
| --- | --- | --- | --- | --- |
| Heinz S, Benner C, Spann N, Bertolino E, Lin YC, Laslo P, Cheng JX, Murre C, Singh H, Glass CK | 2010 | Simple combinations of lineage-determining transcription factors prime cis-regulatory elements required for macrophage and B cell identities | http://www.ncbi.nlm.nih.gov/geo/query/acc.cgi?acc=GSE21512 | Publicly available at the NCBI Gene Expression Omnibus (accession no: GSE21512) |
| Lin YC, Jhunjhunwala S, Benner C, Heinz S, Welinder E, Mansson R, Sigvardsson M, Hagman J, Espinoza CA, Dutkowski J, Ideker T, Glass CK | 2010 | A global network of transcription factors, involving E2A, EBF1 and FOXO1, that orchestrates the B cell fate | http://www.ncbi.nlm.nih.gov/geo/query/acc.cgi?acc=GSE21978 | Publicly available at the NCBI Gene Expression Omnibus (accession no: GSE21978) |
| Hodges E, Molaro A, Dos Santos CO, Thekkat P, Song Q, Uren P, Park J, Butler J, Rafii S, McCombie WR, Smith AD | 2011 | Directional DNA methylation changes and complex intermediate states accompany lineage specificity in the adult hematopoietic compartment | https://www.ncbi.nlm.nih.gov/geo/query/acc.cgi?acc=GSE31971 | Publicly available at the NCBI Gene Expression Omnibus (accession no: GSE31971) |

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
