## [Decision Letter]

Thank you for submitting your article "Tet2 and Tet3 cooperate with B lineage-specific transcription factors to regulate chromatin accessibility" for consideration by *eLife*. Your article has been favorably evaluated by Jessica Tyler as the Senior Editor and three reviewers, one of whom is a member of our Board of Reviewing Editors. The reviewers have opted to remain anonymous.

The reviewers have discussed the reviews with one another and the Reviewing Editor has drafted this decision to help you prepare a revised submission. We understand that the required additional work is extensive and possibly quite expensive, but the reviewers have affirmed the importance of the suggested experiments.

Summary:

Lio et al., studied B cell development in Tet2/3 double knock-out (DKO) mice with genotype *Tet2^-/-^ Tet3^fl/fl^ Mb1cre*. They found a strong block at the pro-B to pre-B cell transition, and a corresponding decrease of mature B cells in the spleen. They also found reduced κ light chain gene rearrangements in DKO pre-B cells, correlating with increased CpG methylation within two enhancers located 3' of C**κ**. Chromatin accessibility measured by ATAC-Seq revealed approximately 6000 sites with reduced accessibility in ex vivo expanded DKO pro-B cells, most of which fell in regions marked with H3K4me1. Tet2/3-regulated enhancers were enriched for binding sites of the transcription factors E2A and PU.1 and shRNA-mediated knockdown of E2A and PU.1 in a BCR-ABL-transformed pre-B cell line resulted in increased methylation at 2 out of 5 sites tested. Partial demethylation was induced at these residues by expressing the Tet2 catalytic domain in a cell line from DKO mice. In addition, loss of Tet2/3 not only impairs DNA demethylation of the κ enhancers but also binding of transcription factor IRF4, suggesting that IRF4 binding at κ enhancers is regulated by Tet2,3 DNA demethylation. They conclude that DNA demethylation by Tet2/3, in part through recruitment by E2A and PU.1, controls κ gene rearrangements in particular and enhancer methylation and accessibility status more generally.

All three reviewers thought the manuscript was interesting and represented a significant advance in defining roles of Tet2/3 in B cell development. All three reviewers also thought that additional mechanistic studies were needed to more fully support some of the major conclusions. The manuscript was viewed to present two parallel themes; one focused on roles of Tet2/3 on κ locus methylation, germline transmission and DNA rearrangement and the other on broader roles of Tet2/3 in controlling the chromatin landscape. With respect to the first theme, the studies of the κ locus do not in themselves explain the developmental block at the pro- to pre-B cell transition. With respect to broader roles of Tet2/3, the authors suggest roles of PU.1 and E2A in recruiting these enzymes to specific genomic locations, but do not examine this possibility directly. In addition, while the manuscript addresses global roles of demethylating enzymes, there is there is no global analysis of differentially methylated regions in the genome.

Essential revisions:

1) Additional experiments are needed to establish that PU.1 and/or E2A function to recruit Tet2/3 to specific genomic targets. These would include; ChIP-seq analysis of PU.1 and E2A in the DKO pro-B cells to demonstrate that these factors can still access their sites in the absence of Tet2 or Tet3. ChIP or ChIP-seq analysis of Tet2 and/or Tet3 in PU.1 and/or E2A knockdown cells should be performed to demonstrate that these factors are required for Tet2/3 recruitment. If Abs directed against Tet2 or Tet3 are an issue, an epitope tagged catalytic domain of Tet2, which is exploited in the complementation analysis, can be used (see point 2). Co-immunoprecipitation experiments should be performed to evaluate whether or not PU.1 and/or E2A form protein complexes with Tet2/3 in solution.

2) ChIP or ChIP-seq experiments are needed to establish that the epitope tagged Tet2 catalytic domain is specifically targeted to the κ enhancers. The ability of the Tet2 catalytic domain to drive κ rearrangement should be tested using the assay employed to analyze the consequences of loss of Tet2,3 on κ rearrangement.

3) There is no global analysis of differentially methylated regions in the genome, though the paper deals with de-methylating enzymes. Instead, ATAC seq is used as a means of assessing the ability of transcription factors to establish regions of open chromatin. However, increases or decreases in ATAC seq peaks could result from effects of the Tet2/3 DKO that are not due to local changes in DNA methylation (e.g., changes in expression of transcription factors). Thus, a global analysis of DNA methylation in the Tet2/3 DKO would substantially strengthen the relationship between DNA methylation, ATAC sensitivity and PU.1/E2A and contribute to an improved understanding of the developmental block.

---

## [Author Response]

*[…] All three reviewers thought the manuscript was interesting and represented a significant advance in defining roles of Tet2/3 in B cell development. All three reviewers also thought that additional mechanistic studies were needed to more fully support some of the major conclusions. The manuscript was viewed to present two parallel themes; one focused on roles of Tet2/3 on κ locus methylation, germline transmission and DNA rearrangement and the other on broader roles of Tet2/3 in controlling the chromatin landscape. With respect to the first theme, the studies of the κ locus do not in themselves explain the developmental block at the pro- to pre-B cell transition. With respect to broader roles of Tet2/3, the authors suggest roles of PU.1 and E2A in recruiting these enzymes to specific genomic locations, but do not examine this possibility directly. In addition, while the manuscript addresses global roles of demethylating enzymes, there is there is no global analysis of differentially methylated regions in the genome.*

*Essential revisions:*

*1) Additional experiments are needed to establish that PU.1 and/or E2A function to recruit Tet2/3 to specific genomic targets. These would include; ChIP-seq analysis of PU.1 and E2A in the DKO pro-B cells to demonstrate that these factors can still access their sites in the absence of Tet2 or Tet3.*

We have performed ChIP-seq analysis for PU.1 in WT and *Tet2/3* DKO pro-B cells with two biological replicates each (new Figure 6—figure supplement 2). Consistent with our model that PU.1 as a pioneer transcription factor to recruit Tet proteins, the global PU.1 binding patterns are virtually indistinguishable between WT and DKO. Of >46,600 peaks identified from both genotypes, only 18 and 4 peaks were selectively occupied (with adj. p ≤ 0.01) in WT and DKO, respectively. Upon examination, none of these 22 peaks were near the Ig**κ** locus or other loci known to be involved in Igκ recombination (new [Supplementary-material SD2-data]). Thus, we conclude that the loss of TET proteins has a very minor effect, if any, on PU.1 binding. Unfortunately, we encountered technical difficulties in performing E2A ChIP and the results were inconclusive.

*ChIP or ChIP-seq analysis of Tet2 and/or Tet3 in PU.1 and/or E2A knockdown cells should be performed to demonstrate that these factors are required for Tet2/3 recruitment. If Abs directed against Tet2 or Tet3 are an issue, an epitope tagged catalytic domain of Tet2 which is exploited in the complementation analysis can be used (see point 2).*

We appreciate the excellent points brought up by the reviewers. We performed ChIP-qPCR for endogenous Tet2 and for ectopically expressed Tet2CD and Tet2CD HxD mutant. All three proteins associate with 3’Eκ and dEκ (new Figure 7). We then analyzed the binding of endogenous Tet2 at these enhancers in PU.1- and E2A-depleted cells and found that Tet2 binding to dEκ was substantially decreased when either transcription factor was depleted with shRNAs (new Figure 6), consistent with increased DNA methylation at this enhancer upon PU.1 or E2A knockdown (Figure 6).

*Co-immunoprecipitation experiments should be performed to evaluate whether or not PU.1 and/or E2A form protein complexes with Tet2/3 in solution.*

We have performed co-immunoprecipitation of endogenous PU.1 or E2A with Tet2 in Abl-transformed pre-B cells and have showed that Tet2 directly associates with both PU.1 and E2A (new Figure 6). To exclude potential indirect interactions due to the presence of contaminating nucleic acids, the experiments were performed in the presence of ethidium bromide and benzonase, a DNA intercalator and a nuclease that cleaves both DNA and RNA, respectively.

*2) ChIP or ChIP-seq experiments are needed to establish that the epitope tagged Tet2 catalytic domain is specifically targeted to the κ enhancers. The ability of the Tet2 catalytic domain to drive κ rearrangement should be tested using the assay employed to analyze the consequences of loss of Tet2,3 on κ rearrangement.*

As mentioned above, we found that ectopically expressed Tet2 catalytic domain (Tet2CD) and its catalytically inactive mutant, Tet2CD HxD, were well expressed in *Tet2/3* DKO pro-B cells (Western blotting, Figure 8) and associated with the 3’Eκ and dEκ enhancers (ChIP-qPCR, new Figure 7). However only Tet2CD, but not Tet2CD HxD, was able to induce Igκ rearrangement and restore chromatin accessibility at the Ig**κ** locus (Figure 7), demonstrating that Tet2 catalytic activity is required and that the decrease in chromatin accessibility and Igκ rearrangement in *Tet2/3* DKO pro-B cells is a reversible phenomenon.

*3) There is no global analysis of differentially methylated regions in the genome, though the paper deals with de-methylating enzymes. Instead, ATAC seq is used as a means of assessing the ability of transcription factors to establish regions of open chromatin. However, increases or decreases in ATAC seq peaks could result from effects of the Tet2/3 DKO that are not due to local changes in DNA methylation (e.g., changes in expression of transcription factors). Thus, a global analysis of DNA methylation in the Tet2/3 DKO would substantially strengthen the relationship between DNA methylation, ATAC sensitivity and PU.1/E2A and contribute to an improved understanding of the developmental block.*

As recommended by the reviewers, we performed whole-genome bisulfite sequencing (WGBS) for WT and *Tet2/3* DKO pro-B cells (two replicates each) and the data are shown in the new Figure 5. In general, as opposed to global changes, loss of *Tet2/3* resulted in specific localized changes of DNA methylation, with 872 and 258 regions with increased and decreased methylation, respectively (new Figure 5).

In interpreting these data, it should be noted that bisulfite sequencing does not distinguish between 5mC and 5hmC. Given that the regions that lose accessibility in *Tet2/3* DKO compared to WT are enriched in 5hmC in WT cells (Figure 4), and that 5hmC levels are dramatically decreased in DKO cells (new Figure 3—figure supplement 2), the increase in 5mC+5hmC observed at the 872 regions in DKO cells compared to WT underestimate the real magnitude of the increase in methylation (5mC). Nonetheless, our data show clearly that the regions with increased methylation in *Tet2/3* DKO pro-B cells are concomitantly less accessible (new Figure 5), suggesting that the decreased chromatin accessibility in DKO cells correlates both with loss of 5hmC and with increased DNA methylation as expected. To further demonstrate the direct causal relationship between TET proteins and accessibility, expression of Tet2CD, but not Tet2CD HxD, was able to restore accessibility to Ig**κ** enhancers and globally at other regions (Figure 7). These data strongly suggest TET protein activity (hydroxymethylation followed by demethylation) facilitates chromatin accessibility, although it has yet to be established whether loss of 5hmC or increased 5mC plays a more significant role in decreased chromatin accessibility in *Tet2/3* DKO pro-B cells.

Regarding the relationship between transcription factors, TET proteins, and chromatin accessibility, we performed ChIP-seq analysis for PU.1 as mentioned above. The overall binding patterns of PU.1 are virtually indistinguishable between WT and DKO (new Figure 6—figure supplement 2). Given that PU.1 directly binds TET2 (new Figure 6), and PU.1 depletion results in lower TET2 binding to dEκ (new Figure 6), we propose that PU.1 first binds to less-accessible nucleosomal DNA and recruits TET2/3, which then remodels the dEκ enhancer for further recruitment of additional transcription factors and/or chromatin regulators. For the convenience of the reader, this hypothesis is now depicted schematically in the new Figure 8.